# AUGMENTED BAYESIAN POLICY SEARCH

**Mahdi Kallel[1]\*, Debabrota Basu[2], Riad Akrour[2], Carlo D'Eramo[1,3,4]**
[1]Center for Artificial Intelligence and Data Science, University of Würzburg, Germany
[2]Univ. Lille, Inria, CNRS, Centrale Lille, UMR 9189 – CRIStAL, Lille, France
[3]Department of Computer Science, TU Darmstadt, Germany
[4]Hessian Center for Artificial Intelligence (Hessian.ai), Germany

## ABSTRACT

Deterministic policies are often preferred over stochastic ones when implemented on physical systems. They can prevent erratic and harmful behaviors while being easier to implement and interpret. However, in practice, exploration is largely performed by stochastic policies. First-order Bayesian Optimization (BO) methods offer a principled way of performing exploration using deterministic policies. This is done through a learned probabilistic model of the objective function and its gradient. Nonetheless, such approaches treat policy search as a black-box problem, and thus, neglect the reinforcement learning nature of the problem. In this work, we leverage the performance difference lemma to introduce a novel mean function for the probabilistic model. This results in augmenting BO methods with the action-value function. Hence, we call our method Augmented Bayesian Search (ABS). Interestingly, this new mean function enhances the posterior gradient with the deterministic policy gradient, effectively bridging the gap between BO and policy gradient methods. The resulting algorithm combines the convenience of the direct policy search with the scalability of reinforcement learning. We validate ABS on high-dimensional locomotion problems and demonstrate competitive performance compared to existing direct policy search schemes.

## 1 INTRODUCTION

The majority of policy gradient literature in Reinforcement Learning (RL), from traditional methods (Sutton & Barto, 2018) to contemporary actor-critic strategies (Schulman et al., 2017; 2015; Haarnoja et al., 2018), employs stochastic policies for experience gathering. This is typically achieved either by modeling policies as probability distributions or by injecting noise into the actions of deterministic policies (Lillicrap et al., 2015; Fujimoto et al., 2018). Having an algorithm that explores effectively using only deterministic policies is preferable when these policies are deployed on physical systems like robotics. Indeed, deterministic policies can prevent the erratic and potentially damaging behavior of stochastic policies while being easier to implement and interpret.

To this end, Bayesian Optimization (BO) methods (Garnett, 2023) perform search directly in the parameter space. BO has emerged as a powerful tool for the global optimization of black-box functions, demonstrating its effectiveness across diverse landscapes and practical applications, such as parameter tuning for machine learning algorithms (Turner et al., 2021; Cowen-Rivers et al., 2022), robotics (Calandra et al., 2016; Muratore et al., 2021), and RL (Bruno et al., 2013). The strength of BO lies in its two core components: (i) a probabilistic model of the objective function, which usually takes the form of a Gaussian Process (GP) prior, and (ii) a sampling procedure that exploits this model to identify informative samples. However BO methods often struggle as the task dimensionality increases since they require a prohibitive number of samples to build a global model.

Local Bayesian Optimization provides an intriguing solution to this challenge (Akrour et al., 2017; Eriksson et al., 2019; Wang et al., 2020; Müller et al., 2021; Nguyen et al., 2022). By focusing on specific regions within the search space, local BO improves the handling of the high-dimensional spaces by promoting more targeted exploration and exploitation, thus reducing the number of evaluations needed to pinpoint optimal solutions. Consequently, there has been a recent upswing in

---

*Correspondence to `mahdi.kallel@uni-wuerzburg.de`.

efforts to scale BO to high-dimensional problems through the design of local schemes. However, when applied to high-dimensional RL problems, these schemes fall short as they treat the policy search problem as a black-box problem, and thus, use only the information of the policy return. By doing so, they overlook the sequential nature of MDPs and discard potentially useful experience.

In this paper, we introduce a principled solution to this problem by building a novel RL-aware mean function to enhance local BO methods. We leverage the performance difference lemma to inject an action-value function into the GP prior of the objective function, thus, effectively incorporating knowledge of past trajectories into our belief about the return of untested policies. We provide a further theoretical ground for our approach by deriving a new bound on the impact of altering deterministic policies on expected returns for Lipschitz MDPs. Then, we show that the posterior gradient yielded by using our new mean function, corresponds to the deterministic policy gradient. Thus, it bridges the gap between BO approaches and policy gradient methods.

The primary contribution of this work is a novel mean function that enhances GPs with the action-value function. Additionally, we propose a fitness-aware adaptive scheme for aggregating multiple $Q$-function approximators. Integrating these components into the Maximum Probability of Descent (MPD) framework (Nguyen et al., 2022) leads to the development of the Augmented Bayesian Search (ABS) algorithm. ABS effectively unifies policy gradient and BO methods, capitalizing on the scalability and sample-efficiency of RL, while also leveraging the principled exploration and practicality offered by BO methods. We provide empirical evidence on the effectiveness of our novel mean function, demonstrating that ABS surpasses previous BO schemes in high-dimensional MuJoCO locomotion problems (Todorov et al., 2012).

## 2 PRELIMINARIES

### 2.1 REINFORCEMENT LEARNING

We consider Markov Decision Processes (MDPs) (Sutton & Barto, 2018) with a continuous bounded state space $\mathcal{S} \subset \mathbb{R}^m$, a continuous action space $\mathcal{A} \subseteq \mathbb{R}^d$, stationary transition dynamics with conditional density $p(s_{t+1}|s_t, a_t)$ satisfying the Markov property, an initial state distribution $\iota$, a reward function $r : \mathcal{S} \times \mathcal{A} \to \mathbb{R}$ and a discount factor $\gamma$. We denote by $\pi : \mathcal{S} \to \mathcal{A}$ a deterministic policy mapping states into actions. Throughout this manuscript, we use only deterministic policies.

At each discrete time step $t$, from a given state $s_t \in \mathcal{S}$, the agent takes an action $a_t = \pi(s_t)$, receiving a reward $r(s_t, a_t)$ and the new state of the environment $s'$ according to the dynamics $p(.|s_t, a_t)$. We denote by $P(s \to s', t, \pi)$ the probability of being at state $s'$ after $t$ transitions following policy $\pi$ starting from $s$. We also denote $\rho_s^\pi(s') \triangleq \sum_{t=0}^\infty \gamma^t P(s \to s', t, \pi)$ the improper discounted state visitation density of $s'$ starting from $s$. By integrating over the initial state distribution $\iota$, we can deduce the improper discounted visitation measure $d^\pi(s') \triangleq \int_\mathcal{S} \rho_s^\pi(s')\iota(s)ds$.

The *action-value function* $Q^\pi$ describes the expected return after taking action $a$ in the state $s$ and thereafter following policy $\pi$, i.e., $Q^\pi(s, a) \triangleq \mathbb{E}_{s' \sim \rho_s^\pi, a'=\pi(s')}\left[r(s', a')|a_0 = a\right]$. Given the action-value function, we can derive the advantage function as $A^\pi(s, a) \triangleq Q^\pi(s, a) - Q^\pi(s, \pi(s))$. The goal of RL is to optimize a policy $\pi_\theta$, parameterized by $\theta$, with the goal of maximizing the *expected discounted policy return* $J(\pi_\theta) \triangleq \mathbb{E}_{s \sim d^{\pi_\theta}, a=\pi_\theta(s)}\left[r(s, a)\right] = \mathbb{E}_{s \sim \iota}\left[Q^{\pi_\theta}(s, \pi(s))\right]$.

To this end, a popular class of RL methods, known as actor-critic, leverages an additional parametric approximation of the action-value function $Q_\phi^{\pi_\theta}$ (Schulman et al., 2015; Haarnoja et al., 2018). When restricting actor-critic methods to deterministic policies, a policy $\pi_\theta$ can be updated using the *deterministic policy gradient*, which changes its actions to maximize $Q^{\pi_\theta}$ (Silver et al., 2014):

$$\nabla_\theta J(\pi_\theta) = \mathbb{E}_{s \sim d^{\pi_\theta}}\left[\nabla_a Q^{\pi_\theta}(s, a)\Big|_{a=\pi_\theta(s)} \nabla_\theta \pi_\theta(s)\right]. \tag{1}$$

### 2.2 INFORMATION MAXIMIZING BAYESIAN OPTIMIZATION

Bayesian Optimization (BO) is a sequential method for global optimization of black-box functions when sample-efficiency is paramount. It builds a probabilistic model of the objective function, often a Gaussian Process (GP), which is used by an acquisition function that guides the parameter

space exploration. However, BO struggles with high-dimensional spaces as it requires a prohibitive amount of data to build a global model of the objective function. To address these issues, local BO methods have been developed (Nguyen et al., 2022; Müller et al., 2021; Eriksson et al., 2019; Fröhlich et al., 2021). These methods constrain the search to a subspace of interest, circumventing difficulty of modeling and finding promising candidates in high dimensional problems.

A recent development in the local BO methods is the introduction of Information Maximizing BO. First introduced by Müller et al. (2021), and subsequently refined by Nguyen et al. (2022), these methods rely on an acquisition function that seeks to maximize a local measure of information. By considering a **GP** belief about the objective function $J \sim \text{GP}\left(m(x), K(x, x')\right)$ with a differentiable mean function $m$ and a twice-differentiable covariance function $K$, we have that the joint distribution between the **GP** and its derivative is still a **GP** (Rasmussen, 2003). Hence, by conditioning on a dataset of observations $(X, Y)$, the posterior distribution of the derivative at a point of interest $\theta$ takes the form of a Gaussian distribution $p\left(\nabla J(\theta) \mid \theta, X, Y\right) = \mathcal{N}(\mu_\theta, \Sigma_\theta)$, where

$$\mu_\theta \triangleq \nabla_\theta m(\theta) + \nabla_\theta K(\theta, X) K(X, X)^{-1}\left(Y - m(X)\right), \tag{2}$$

$$\Sigma_\theta \triangleq \nabla_\theta K(\theta, \theta) - \nabla_\theta K(\theta, X) K(X, X)^{-1} \nabla_\theta K(X, \theta). \tag{3}$$

In Nguyen et al. (2022), the measure of information is taken to be the probability of descent at a **central** point $\theta$. Subsequently, an acquisition function $\alpha(z|\theta, X, Y)$ is developed to guide the exploration towards an **acquisition** point $z$, such that adding $(z, y_z)$ to our observation dataset maximizes the probability of descent at $\theta$. The resulting BO algorithm, named MPD (Nguyen et al., 2022), alternates between two steps. Starting from a **central** point $\theta$, it uses the acquisition function $\alpha$ to *sequentially* query multiple **acquisition** points $z$ and observe their values $y_z$ until obtaining satisfying information about $\theta$. Then, MPD uses the better-estimated gradient $\mu_\theta$ to move along the most probable descent direction $\nu_\theta \triangleq \Sigma_\theta^{-1} \mu_\theta$ to a new point $\theta'$, and repeats this loop.

## 3 REINFORCEMENT LEARNING-AWARE BAYESIAN OPTIMIZATION

The mean function of a GP determines the expected value at a given point. In *interpolating* regions, the posterior mean is largely influenced by observed data points due to significant correlation. Conversely, in *extrapolating* regions, the data's influence is minimal, causing the posterior mean to revert to the mean function. Despite its importance, this function has received little attention in BO literature, especially in RL applications where uninformative priors like constant functions are favored. In this section, we leverage the MDP properties of the problem to build a better mean function.

### 3.1 ON THE SMOOTHNESS OF DETERMINISTIC POLICY RETURNS

First, we develop a new bound on the impact of altering deterministic policies on expected returns in the particular case of Lipschitz MDPs (Pirotta et al., 2015). We assume that the implementation of a policy $\pi_\theta$ gives access to $J(\pi_\theta)$, $d^{\pi_\theta}$, and $Q^{\pi_\theta}$. Given an alternative policy $\pi_x$, our objective is to accurately estimate its return $J(\pi_x)$ [1]. The **performance difference lemma** (Kakade & Langford, 2002) can establish the relation between deterministic policies returns and the advantage function :

$$J(\pi_x) - J(\pi_\theta) = \mathbb{E}_{s \sim d^{\pi_x}}\left[A^{\pi_\theta}(s, \pi_x(s))\right] = \langle d^{\pi_x}(.), A^{\pi_\theta}(., \pi_x(.))\rangle. \tag{4}$$

Direct application of this lemma to infer $J(\pi_x)$ is not feasible due to the need for $d^{\pi_x}$. However, by reordering terms, we can express policy return as an estimable term plus an unknown residual.

$$J(\pi_x) = \underbrace{J(\pi_\theta) + \langle d^{\pi_\theta}(.), A^{\pi_\theta}(., \pi_x(.))\rangle}_{\text{Estimate}} + \underbrace{\langle (d^{\pi_x} - d^{\pi_\theta})(.), A^{\pi_\theta}(., \pi_x(.))\rangle}_{\text{Residual}}. \tag{5}$$

The following assumptions allow us to bind the residual term while using deterministic policies.

**Assumption 3.1.** *An MDP is $(L_r, L_p)$-Lipschitz if for all $s, s' \in \mathcal{S}$ and $a, a' \in \mathcal{A}$:*

- $|r(s, a) - r(s', a')| \leq L_r(d_\mathcal{S}(s, s') + d_\mathcal{A}(a, a'))$,
- $W(p(.|s, a), p(.|s', a')) \leq L_p(d_\mathcal{S}(s, s') + d_\mathcal{A}(a, a'))$.

---

[1]The findings of this section are relevant to both parametric and non-parametric policies. Parametric notations are used for consistency throughout the paper.

**Assumption 3.2.** *A policy $\pi_x$ is $L_\pi$-Lipschitz if for all $s, s' \in \mathcal{S}$ :*

- $W(\pi_x(s), \pi_x(s')) \le L_\pi(d_\mathcal{S}(s, s'))$.

Here, we denote by $d_\mathcal{S}$, $d_\mathcal{A}$ the distances in $\mathcal{S}$ and $\mathcal{A}$ respectively. In this work, we take this distance to be the Euclidean distance $\|.\|$ and $W$ to be the Wassertstein-2 distance on the space of measures.

**Theorem 3.3.** *For an $(L_r, L_p)$-Lipschitz MDP operating with deterministic $L_\pi$-Lipschitz policies, and $\gamma L_p(1 + L_\pi) < 1$, we bound the residual term for any policies $\pi_x$ and $\pi_\theta$ as*

$$|\langle d^{\pi_x} - d^{\pi_\theta}, A^{\pi_\theta}(.|\pi_x(.))\rangle| \le C \times \sup_s \|\pi_x(s) - \pi_\theta(s)\|, \quad where \; C \triangleq \frac{2\gamma L_\pi L_r (1 + L_\pi)}{(1 - \gamma L_p(1 + L_\pi))^2}. \quad (6)$$

The bound describes the worst-case value of the residual term. Theorem 3.3 extends the existing bounds on the residual term (Kakade & Langford, 2002; Schulman et al., 2015) to our setting, where both the policy and the dynamics are deterministic. Similarly, we extend this bound to the case of linear policies and bounded state spaces as

$$|\langle d^{\pi_x} - d^{\pi_\theta}, A^{\pi_\theta}(.|\pi_x(.))\rangle| \le C \times \sup_s \|s\|\|x - \theta\|. \quad (7)$$

## 3.2 ADVANTAGE MEAN FUNCTION

Now, we construct a GP that models the policy return $J$ over the space of parameters of deterministic *linear* policies. To solve the policy search problem, we can query an oracle to obtain a noisy evaluation $\widehat{J}(\pi_x) = J(\pi_x) + \omega$ of **any** point $x$, an empirical estimate of its discounted state distribution $\widehat{d^{\pi_x}} \simeq d^{\pi_x}$, and the transitions of the corresponding trajectory. In order to construct such a model, first, we roll out a **central** point $\theta$ and obtain estimates $\widehat{J}(\pi_\theta)$ and $\widehat{d^{\pi_\theta}}$. Then, we train a deep neural network parameterized by $\phi$ to estimate the action-value function $\widehat{Q}_\phi^{\pi_\theta} \simeq Q^{\pi_\theta}$. For **any** policy $\pi_x$ parameterized by $x$, if $\sup_s \|x - \theta\|$ is small enough, Theorem 3.3 and Equation (7) indicate that a good prior on its return $J(\pi_x)$ is an estimate of the first term in Equation (5), i.e.

$$\widehat{m}_\phi(x) \triangleq \widehat{J}(\pi_\theta) + \mathbb{E}_{s \sim \widehat{d^{\pi_\theta}}} \left[ \widehat{A}^{\pi_\theta}(s, \pi_x(s)) \right]. \quad (8)$$

Therefore, we advocate using $\widehat{m}_\phi$ as a mean function for our GP, which we call *Advantage Mean Function*. We emphasize that the advantage mean function $\widehat{m}_\phi(x)$ for the policy return $J(\pi_x)$ depends on the current **central** point $\theta$. By using $\widehat{m}_\phi$, we leave the GP to model the residual term from Equation (5), which can be viewed as a second-order term (Saleh et al., 2022), and the errors due to empirical estimates and approximations. This is in contrast to the typical constant mean function, which forces the GP to fit all the variations of the objective function. By plugging in the advantage mean function in the posterior of the derivative $\mu_\theta$ (Equation (2)), we unravel an interesting property.

**Corollary 3.3.1.** *Given the mean function $\widehat{m}_\phi(\cdot)$, the mean of the gradient posterior at $\theta$ is*

$$\mu_\theta = \mathbb{E}_{s \sim \widehat{d^{\pi_\theta}}} \left[ \nabla_a \widehat{Q}_\phi^{\pi_\theta}(s, a) \Big|_{a=\pi(s)} \nabla_\theta \pi_\theta(s) \right] + \nabla_\theta K(\theta, X) \, K(X, X)^{-1}(\widehat{J}(X) - \widehat{m}_\phi(X)), \quad (9)$$

*where $X \triangleq \{z_1, z_2, \ldots, \} \cup \{\theta_1, \theta_2, \ldots\}$ denotes the set of policy parameters of the past observations, $\widehat{J}(X) = \{\widehat{J}(\pi_{x_1}), \widehat{J}(\pi_{x_2}), \ldots\}$ denotes the set of their noisy policy evaluations, and $\widehat{m}_\phi(X) = \{\widehat{m}_\phi(x_1), \widehat{m}_\phi(x_2), \ldots\}$ denotes the mean function estimate of their policy return.*

The mean of the gradient posterior, $\mu_\theta$, takes the form of the Deterministic Policy Gradient (Silver et al., 2014), corrected by a factor that is proportional to the alignment of the advantage mean function with the observed data points. Given that our objective function is the policy return, the mean of the gradient posterior aims to get a closer correspondence with the actual deterministic policy gradient, as compared to the estimate obtained by using only the estimator $\widehat{Q}_\phi^{\pi_\theta}$.

In Figure 1, we demonstrate the behavior of the acquisition function of MPD (Nguyen et al., 2022). We observe its tendency to keep the **acquisition** points $z$ in the immediate vicinity of the **central** point $\theta$ effectively controlling $\|z - \theta\|$ and therefore the residual term of Equation (5). In conclusion, the proposed mean function $\widehat{m}_\phi$ integrates both zero-order (returns) and first-order (gradient) observations into the BO framework, as hypothesized in (Müller et al., 2021), thereby seamlessly bridging the gap between policy gradient and Bayesian Optimization methods.

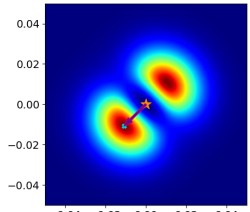 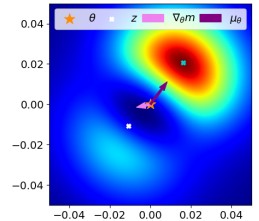 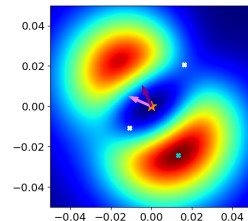

(a) $t = 0$: Observed only $\theta$ (star)  (b) $t = 1$: Observed $\theta$, $z_1$ (white)  (c) $t = 2$: Observed $\theta$, $z_1$, $z_2$ (white)

Figure 1: Behavior of the acquisition function of MPD (Nguyen et al., 2022) augmented with the advantage mean function (Equation (8)). The maximum of the acquisition function (blue dot) lies in the direction of the mean of the gradient posterior at $\theta$ (star), i.e. $\mu_\theta$ (violet line). The posterior corrects the mean gradient $\nabla_\theta \widehat{m}_\phi$ (pink line) when the mean function $\widehat{m}_\phi$ does not fit the observations.

# 4 ENHANCING $Q$-FUNCTION ESTIMATORS: EVALUATION & AGGREGATION

The mean function, depicted in Figure 1, steers the acquisition function in its search for points that maximize the ascent probability at the central point $\theta$. Equation (2) indicates that the same function influences the computation of the descent direction. Consequently, the quality of the approximator $\widehat{Q}_\phi^{\pi_\theta}$, embedded in the advantage mean function, is crucial for both exploration and exploitation. Therefore, it is desirable for $\widehat{Q}_\phi^{\pi_\theta}$ to generalize beyond observed trajectories. In this section, we propose a criterion for estimating the quality of such approximators. Utilizing this criterion, we further develop an adaptive strategy for aggregating the predictions of an ensemble of approximators.

## 4.1 EVALUATING $Q$-FUNCTION ESTIMATORS

In the existing literature (Fujimoto et al., 2018; Van Hasselt et al., 2016), the quality of a parametric approximator $\widehat{Q}_\phi^{\pi_\theta}$ is assessed by comparing its predictions to those obtained by an empirical estimate $\widehat{Q}^{\pi_\theta}$ generated by rolling out the policy $\pi_\theta$. The weakness of this scheme is that it only uses the trajectories collected by $\pi_\theta$, which cover only a subset of the state-action space.

To tackle this limitation, we leverage the performance difference lemma (Equation (4)). This property indicates that given an action-value function $Q^{\pi_\theta}$ and the corresponding policy return $J(\pi_\theta)$, we can perfectly recover the policy return $J(\pi_x)$ of **any** actor $\pi_x$, if $d^{\pi_x}$ is also known.

In our setting, we roll out the policy corresponding to the **central** point $\theta$ to have access to an empirical estimate $\widehat{J}(\pi_\theta)$ and an approximator $\widehat{Q}_\phi^{\pi_\theta}$. Similarly, we roll out the policy corresponding to **any** point $x$ to get access to the sample estimates of $\widehat{J}(\pi_x)$ and $\widehat{d}^{\pi_x}$. Thanks to the performance difference lemma (Equation (4)), we can use the error induced by $\widehat{Q}_\phi^{\pi_\theta}$ as a measure of its quality :

$$\left( \widehat{J}(\pi_\theta) + \mathbb{E}_{s \sim \widehat{d}^{\pi_x}} \left[ \widehat{A}_\phi^{\pi_\theta}(s, \pi_x(s)) \right] - \widehat{J}(\pi_x) \right)^2 \triangleq \left( \widetilde{m}_\phi(x) - \widehat{J}(\pi_x) \right)^2. \tag{10}$$

We emphasize that $\widehat{m}_\phi$ relies on $\widehat{d}^{\pi_\theta}$, while $\widetilde{m}_\phi$ leverages $\widehat{d}^{\pi_x}$ to estimate the policy return $J(\pi_x)$. This method evaluates $\widehat{Q}_\phi^{\pi_\theta}$ on trajectories from policies other than $\pi_\theta$, allowing all available trajectory data to be used for our approximator's evaluation. This provides a more comprehensive validation criterion, addressing the existing assessment scheme's limitations.

## 4.2 ADAPTIVE AGGREGATION OF $Q$-FUNCTION ESTIMATORS

In our setting, we have access to the dataset $\mathcal{D} = \{X, \widehat{J}(X), \widehat{d}^\pi(X)\}$ collected during the previous iterations of the algorithm. It consists of the observation points $X$, their sample policy returns $\widehat{J}(X)$, and empirical discounted state occupation measures $\widehat{d}^\pi(X) \triangleq \{d^{\pi_{x_1}}, d^{\pi_{x_2}}, \ldots\}$. Assuming a good approximator $\widehat{Q}_\phi^{\pi_\theta}$, Equation (10) suggests $\widetilde{m}_\phi$ should serve as an effective predictor of $J(\pi_x)$.

---

**Algorithm 1** Rollout for one point

---

**Require:** Policy parameters $x$, observation dataset $\mathcal{D} = \{\}$, replay buffer $\mathcal{B} = \{\}$
 1: **if** $x$ is **central then**
 2:     Reset the optimizers of all critics
 3:     Reset the weights of least performing critic
 4: **end if**
 5: Collect $\widehat{J}(\pi_x) = J(\pi_x) + \epsilon, \widehat{d^{\pi_x}}, \widehat{T}(\pi_x) \triangleq \{s_t, a_t, r_t, s_{t+1}\}_{\{t=1,\ldots,H\}}$
 6: Update training data $\mathcal{D} \leftarrow \mathcal{D} \cup (x, \widehat{J}(\pi_x), \widehat{d^{\pi_x}})$ replay buffer $\mathcal{B} \leftarrow \mathcal{B} \cup \widehat{T}(\pi_x)$
 7: Update the parameters of critics $\widehat{Q}^{\pi_\theta}_{\phi=\{\phi_1,\ldots,\phi_n\}}$ on $\mathcal{B}$
 8: Compute the weights of every critic according to Equation (12) using $\mathcal{D}$
 9: Fit the GP parameters on $\mathcal{D}$ and $\widehat{m}_\Phi$.

---

Therefore, to evaluate the quality of $\widehat{Q}^{\pi_\theta}_\phi$, we use the coefficient of determination of $\widetilde{m}_\phi$ on the dataset $\mathcal{D}$. This metric measures the percentage of variance in the data explained by our predictor :

$$\widetilde{R}^2(\phi|\theta, \mathcal{D}) \triangleq 1 - \frac{\sum_{x \in X}\left(\widetilde{m}_\phi(x) - \widehat{J}(\pi_x)\right)^2}{\sum_{x \in X}\left(\bar{J}(X) - \widehat{J}(\pi_x)\right)^2} \quad \text{where} \quad \bar{J}(X) \triangleq \frac{\sum_{x \in X}\widehat{J}(\pi_x)}{|X|}. \tag{11}$$

Estimating $Q^{\pi_\theta}(s, a)$ requires rolling out trajectories $\tau(s, a; \pi_\theta) \triangleq \{s, a, s', \pi_\theta(s'), s'', \pi_\theta(s''), \ldots\}$. These trajectories play the role of the training data to learn a good approximation $Q^{\pi_\theta}_\phi(s, a)$. Hence, we refer to $\widetilde{R}^2(\phi|\theta, \mathcal{D})$ as the *validation score* since the trajectories $\tau(s, \pi_x(s); \pi_\theta)$ for $s \sim d^{\pi_x}$ used by $\widetilde{m}_\phi$ are not present in our training dataset. Similarly, we can define $\widehat{R}^2(\phi|\theta, \mathcal{D})$ for the predictor $\widehat{m}_\phi$. The trajectories $\tau(s, \pi_x(s); \pi_\theta)$ for $s \sim d^{\pi_\theta}$ used by $\widehat{m}_\phi$ are different from the validation trajectories. Since $\widehat{m}_\phi$ is used during inference, we refer to $\widehat{R}^2(\phi|\theta, \mathcal{D})$ as the *test score*. We note that while a perfect $Q^{\pi_\theta}$ estimator can have a perfect validation score, it might not achieve a perfect test score due to the residual term in Equation (5).

In our context, we deploy an ensemble of critics $\widehat{Q}^{\pi_\theta}_{\{\phi_1,\ldots,\phi_n\}}$. We adopt the Follow The Regularised Leader (FTRL) algorithm (Cesa-Bianchi & Lugosi, 2006) to construct an aggregated critic, $\widehat{Q}^{\pi_\theta}_\Phi$, via softmax weighting of the predictions of each critic using its respective $\widetilde{R}^2$ score:

$$\widehat{Q}^{\pi_\theta}_\Phi(s, a) = \sum_{i=1}^n w_i \widehat{Q}^{\pi_\theta}_{\phi_i}(s, a) \qquad \text{where} \qquad w_i \triangleq \frac{\exp \widetilde{R}^2(\phi_i|\theta, \mathcal{D})}{\sum_{j=1}^n \exp \widetilde{R}^2(\phi_j|\theta, \mathcal{D})}. \tag{12}$$

We denote by $\widehat{m}_\Phi$ the advantage mean function that uses the aggregate estimate $\widehat{Q}^{\pi_\theta}_\Phi$. Additionally, we reset the optimizer of every critic when changing the **central** point $\theta$ because we are learning the $Q$-function of a new policy. We also reset the weights of the worst-performing critic in order to improve the general performance by avoiding the primacy bias (Nikishin et al., 2022). This resetting also helps in decorrelating the predictions of each critic, and thus, reducing the variance of the predictions. We illustrate this step in Algorithm 1 and describe the full BO scheme in Algorithm 2.[2]

---

**Algorithm 2** Augmented Bayesian Search (ABS)

---

**Require:** central point $\theta$, number of iterations $N$, number of acquisition points $M$, stepsize $\delta$, observation dataset $\mathcal{D} = \{\}$, replay buffer $\mathcal{B} = \{\}$
 1: **for** $i = 0, \ldots, N$ **do**
 2:     Rollout the **central** point $\theta$ using Algorithm 1.
 3:     **for** $j = 0, \ldots, M$ **do**
 4:         Query for an **acquisition** point $z = \arg\max_z \alpha(z|\theta, \mathcal{D})$
 5:         Rollout the **acquisition** point $z$ using Algorithm 1.
 6:     **end for**
 7: **end for**
 8: Move in the direction of the most probable descent $\theta \leftarrow \theta + \delta \times \Sigma_\theta^{-1}\mu_\theta$

---

[2]The main BO loop of ABS can be seen as a simplified version of that of MPD. The only difference being that MPD performs multiple descent steps and uses a constant mean, whereas our method uses the advantage mean $\widehat{m}_\phi$, which depends on the current **central** point $\theta$ limiting us to one descent step.

## 5 EXPERIMENTAL ANALYSIS

In this experimental analysis, we aim to answer the following questions: **(I)** Is our advantage mean function a good prior on the policy return? **(II)** Does our advantage mean function improve the efficiency of BO methods? **(III)** What role does our adaptive aggregation play?

### 5.1 IMPLEMENTATION DETAILS

For our $Q$-function estimator, we leverage the state-of-the-art ensemble method of DroQ networks (Hiraoka et al., 2021) with default parameters. For our ensemble of critics we use 5 distinct DroQ networks. After querying for an **acquisition** point $z$, we perform 5000 gradient steps to the $Q$-function. For our covariance function we choose a squared exponential kernel $SE(x, x') = \sigma_f^2 \exp \sum_{i \leq d} \frac{\|x-x'\|_2^2}{\sigma_i^2} + \sigma_n^2 \delta_{x=x'}$, where $\sigma_f, \sigma_n, \sigma_{i \in \{1,...,d\}}$ are the signal variance, the observation noise, and lengthscales for dimension $i$, respectively. We model the policy return as a GP over the parameters of deterministic linear policies $J(x) \sim GP(m_\phi(x), SE(x, x'))$.

BO requires prior knowledge about the tasks in order to fix the hyperpriors for parameters, such as the signal variance $\sigma_f$ and the observation noise $\sigma_n$, adding more hyperparameters to the problem. For our implementation of MPD and ABS, we rely on a heuristic to derive dynamic hyperpriors for $\sigma_f, \sigma_n$ from the observation data. This leaves us only to set the lengthscale hyperprior alleviating the burden of tuning these parameters. We refer the reader to the Appendix A for further details.

We validate our contributions using MuJoCo locomotion tasks (Todorov et al., 2012). We use a discount factor $\gamma = 0.99$ for all tasks, apart from Swimmer where it is $\gamma = 0.995$. For our experiments, we use the state normalization scheme employed in (Mania et al., 2018; Nguyen et al., 2022) where the states are normalized to a normal distribution using online estimates of the mean and the variance. We have implemented ABS using Python 3.10 and Jax 0.4.20, we run all of our experiments on a cluster of NVIDIA A100 40 GB GPU. Our code is available at `https://github.com/kallel-mahdi/abs`.

### 5.2 RESULTS AND ANALYSIS

**Goodness of Fit of the Advantage Mean Function.** In Figure 2, we show the evolution of the **validation** metric of the best critic in the ensemble $\widetilde{R}^2$, taking all the samples collected in the last 3 outer loop steps of Algorithm 2. We superpose to it the **test** metric $\widehat{R}^2$ of the ensemble $\widehat{Q}_\Phi^{\pi_\theta}$ over the acquisition points sampled during the same step. This is done for 3 high-dimensional tasks of MuJoCo (Todorov et al., 2012). We observe that the **validation** metric $\widetilde{R}^2$ remains consistently positive with the validation predictor $\widetilde{m}_\Phi$ explaining on average approximately $50\%$ of the variance observed data. The **test** metric $\widehat{R}^2$ manages also to be positive quite frequently, while being sometimes negative, meaning that our mean function $\widehat{m}_\phi$ fails sometimes to explain the variance of policy return for the acquisition points. However, this is not a big concern as we only need the mean function to be able to identify one good direction of descent at a time and not be a perfect predictor for every point. Similar to a validation and test error, we observe a positive correlation between both metrics and that the **validation** error acts like an upper bound to its **test** counterpart as in the supervised setting.

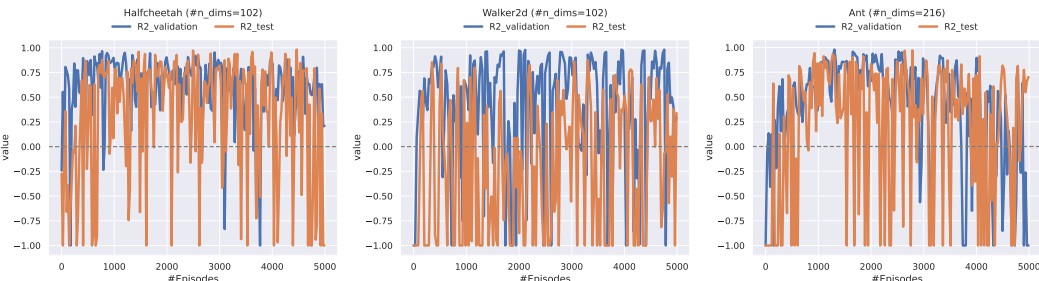

Figure 2: Evolution of the validation and test scores on some of the MuJoCo tasks. We plot the results of a seed to facilitate the interpretation of our results. We provide the histogram and correlations of these distributions in the Appendix.

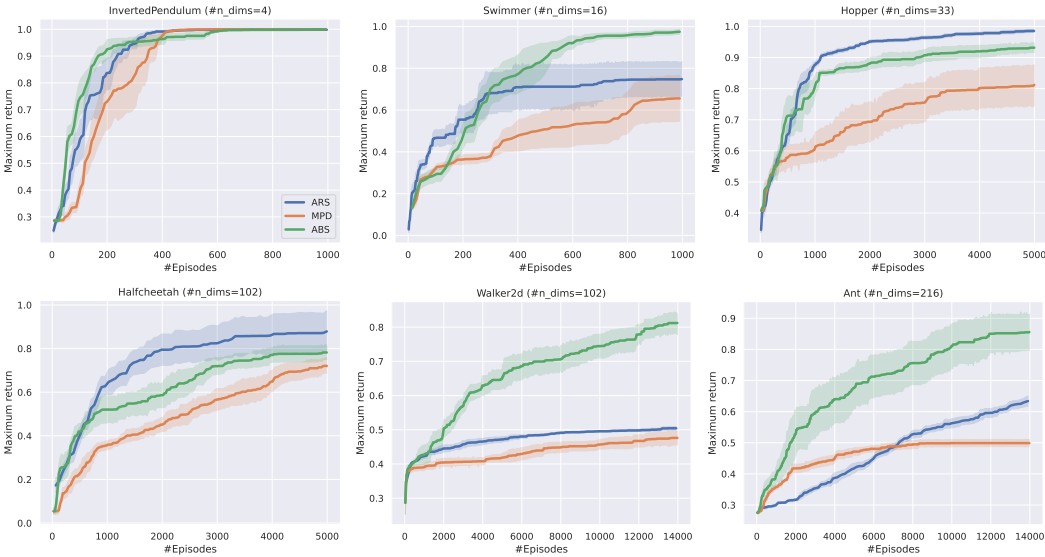

Figure 3: Evolution of the maximum discounted policy return on the MuJoCo-v4 tasks. We use 5 random seeds for every algorithm. We report the undiscounted returns in Figure 6 in the Appendix.

**Performance of ABS in terms of Policy Return.** We evaluate ABS against two baselines, namely, MPD which is the current state-of-art of local BO methods (Nguyen et al., 2022) and Augmented Random Search (ARS) (Mania et al., 2018), an evolutionary algorithm that estimates the gradient of the noisy objective using random perturbations of the policy. In Figure 3, we see that ABS is competitive with MPD and ARS in low-dimensional tasks. Notably, ABS consistently outperforms the MPD demonstrating that the advantage mean function is helpful for local BO schemes. The relative performance of ABS improves in high-dimensional tasks, such as Walker2d and Ant, where the curse of dimensionality makes it harder for ARS and MPD to identify promising descent directions, whereas our mean function can focus the search on more promising regions of the parameter space.

**Effectiveness of the Aggregation Scheme.** We perform an ablation study to confirm the effectiveness of our adaptive scheme for aggregating critics in Figure 4. First, we observe that an ensemble of critics outperforms a single critic. We also notice that our strategy of adaptive aggregation (Equation 12) outperforms average ensembling. This is especially in the initial phases when transition data is scarce and it is easy to have a poorly performing critic. Resetting the least performing critic improves the performance of the ensemble, and its effect is more pronounced when it is combined with our adaptive aggregation method. We explain this performance by the capability of our **validation** metric to detect when the recently reset critic performs poorly and ignore its predictions.

To summarize, ABS, which can be viewed as MPD augmented with the advantage mean function and the adaptive aggregation scheme consistently outperforms the original MPD that uses the uninformative constant mean function. The advantage mean function can explain a fair portion of the policy return of the observation points and hence represents a good prior. Our validation metric and the consequent aggregation scheme are effective in dynamically selecting good critics.

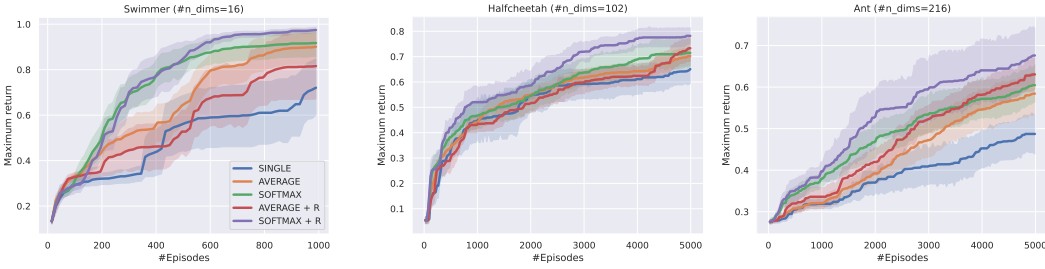

Figure 4: Ablation study: Effect of the adaptive aggregation on the performance of ABS. Combining adaptive aggregation and resetting the worst critic outperforms all baselines.

## 6 RELATED WORK

Several prior works applied a Bayesian view to reinforcement learning and policy gradient methods (Ghavamzadeh et al., 2016). However, in **Bayesian Optimization**, the majority of works treat reinforcement learning as a black-box problem (Martinez-Cantin et al., 2007; Englert & Toussaint, 2016; Martinez-Cantin, 2018; Eriksson et al., 2019; Nguyen et al., 2022) and are oblivious to the MDP properties. As an exception, Wilson et al. (2014) propose a GP mean function leveraging a dynamics-model estimate as a prior for the policy return and develop a kernel function measuring similarity between policies using trajectory data. In the **black-box setting**, Akrour et al. (2017) use a sampling-based approach where the parameters of a normal distribution are updated using information-theoretic constraints leading to optima. Fröhlich et al. (2021) introduce a confidence region BO scheme to constrain the search space to points with less uncertainty. Eriksson et al. (2019) introduce a trust region BO scheme that uses a collection of simultaneous local optimization runs using independent probabilistic models. Müller et al. (2021) propose the Gradient Information BO algorithm (GIBO) that utilizes a probabilistic model of the objective function and its gradient to maximize information about the gradient posterior. Finally, Nguyen et al. (2022) improve on the information criterion of GIBO by querying for points that maximizes the probability of descent, and then moves in the direction of most probable descent. In contrast, our work exploits the sequential nature of the task which provides large practical gains, especially on higher dimensional tasks.

In the **direct policy search** literature, one alternative to BO is the use Particle Swarms (Hein et al., 2016; Hein, 2019). Closer to our work are Evolutionary Strategies (ES). They are gradient-free methods deploying a random process to iteratively generate candidate solutions. Then, they evaluate these solutions and bias the search in the direction of the best-scoring ones (Hansen, 2006; Salimans et al., 2017; Mania et al., 2018). (Salimans et al., 2017) propose a variant of ES for optimizing the policy parameters. They estimate a gradient via Gaussian smoothing of the objective function. (Mania et al., 2018) build on the work of (Salimans et al., 2017) and introduce a simple random search algorithm that is competitive with the state of the art while using only linear policies. Compared to BO, ES uses a less principled exploration strategy and, in comparison to our work, does not include the structure of the MDP in its search.

The **deterministic policy gradient** (Silver et al., 2014) and its practical variants (Lillicrap et al., 2015; Fujimoto et al., 2018) are also closely related to the context of our work. These methods apply the deterministic actor-critic algorithm to learn deep neural network policies. Kakade & Langford (2002); Schulman et al. (2015) develop algorithms for learning monotonously improving policies. This class of algorithm relies on lower bounding the difference of return between the current and the potential next policy and taking a step such that we are sure to improve our policy. Saleh et al. (2022) are the first to derive such bounds for deterministic policies, but we specify these bounds further for Lipschitz MDPs. We further leverage this result to design our advantage mean function.

## 7 CONCLUSION AND DISCUSSION

In this work, we have presented Augmented Bayesian policy Search (ABS), a method that leverages a new mean function for the Gaussian Process (GP) prior which is tailored to Reinforcement Learning (RL). We derive our method by resorting to the performance difference lemma, to inject an action-value function of deterministic policies into the GP prior on the policy return. Our advantage mean function acts like a first order taylor expansion of the policy return, bridging the gap between BO and RL methods. We theoretically ground our approach by deriving a new bound on the impact of altering deterministic policies on the expected returns for Lipschitz MDPs. Moreover, we propose a novel adaptive scheme for aggregating multiple $Q$-function estimators. Empirically, we show that ABS scales to high-dimensional problems and establishes state-of-the-art performance for Bayesian Optimization (BO) methods in MuJoCo locomotion problems. ABS can explore using deterministic policies and learn effectively in an episodic setting. It also allows trading-off compute for learning speed, which is a highly desirable property in real-world applications.

For this work, we were limited to linear policies due to computational considerations. A promising future direction is scaling BO methods to deep policies. To this end, viable solutions include random projection methods (Ziomek & Ammar, 2023) and kernel functions for the GP that are tailored for deterministic policies, close to works like Wilson et al. (2014); Ghavamzadeh et al. (2016).

## ACKNOWLEDGMENTS

The authors thank Gandharv Patil and Tom Dupuis for their helpful comments and discussions. This work was funded by the German Federal Ministry of Education and Research (BMBF) (Project: 01IS22078). This work was also funded by Hessian.ai through the project 'The Third Wave of Artificial Intelligence – 3AI' by the Ministry for Science and Arts of the state of Hessen. D. Basu acknowledges the Inria-Kyoto University Associate Team "RELIANT", and the ANR JCJC for the REPUBLIC project (ANR-22-CE23-0003-01) for supporting this work. The authors gratefully acknowledge the scientific support and HPC resources provided by the Erlangen National High Performance Computing Center (NHR@FAU) of the Friedrich-Alexander-Universität Erlangen-Nürnberg (FAU) under the NHR project b187cb. NHR funding is provided by federal and Bavarian state authorities. NHR@FAU hardware is partially funded by the German Research Foundation (DFG) – 440719683.

## AUTHOR CONTRIBUTIONS

R. Akrour proposed the idea of the advantage mean function which was explored as part of the internship of M. Kallel at Inria Scool under the supervision of R. Akrour and D. Basu. After his internship, M. Kallel started his Ph.D. at the University of Würzburg under the supervision of C. D'Eramo, where he kept working on the project and finalized it. All the authors contributed to the writing of the paper.

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

## A  IMPLEMENTATION DETAILS

| Task name | Learning rate | Lengthscale prior |
|---|---|---|
| InvertedPendulum | $\{0.0025, 0.005\}^*$ | $\{\mathcal{U}(0.00125, 0.025), \mathcal{U}(0.0025, 0.05)\}$ |
| Swimmer | $\{0.0025, 0.005\}^*$ | $\{\mathcal{U}(0.00125, 0.025), \mathcal{U}(0.0025, 0.05)\}$ |
| Hopper | $\{0.0025, 0.005\}^*$ | $\{\mathcal{U}(0.00125, 0.025), \mathcal{U}(0.0025, 0.05)\}$ |
| Halfcheetah | $\{0.00125, 0.0025\}^*$ | $\{\mathcal{U}(0.000625, 0.0125), \mathcal{U}(0.00125, 0.025)\}$ |
| Walker2d | $\{0.00125, 0.0025\}^*$ | $\{\mathcal{U}(0.000625, 0.0125), \mathcal{U}(0.00125, 0.025)\}$ |
| Ant | $\{0.00125, 0.0025\}^*$ | $\{\mathcal{U}(0.000625, 0.0125), \mathcal{U}(0.00125, 0.025)\}$ |

Table 1: Hyperparameters used for grid-search for MPD and ABS. (*) For MPD we use only a learning rate of $0.01$.

| Task name | Learning rate | Lengthscale prior | $N_c$ | $N_{acq}$ | $N_{max}$ |
|---|---|---|---|---|---|
| InvertedPendulum | 0.005 | $\mathcal{U}(0.0025, 0.05)$ | 2 | 6 | 21 |
| Swimmer | 0.0025 | $\mathcal{U}(0.0025, 0.05)$ | 3 | 12 | 39 |
| Hopper | 0.0025 | $\mathcal{U}(0.0025, 0.05)$ | 3 | 16 | 51 |
| Halfcheetah | 0.0025 | $\mathcal{U}(0.00125, 0.025)$ | 4 | 20 | 63 |
| Walker2d | 0.0025 | $\mathcal{U}(0.000625, 0.0125)$ | 4 | 20 | 63 |
| Ant | 0.0025 | $\mathcal{U}(0.000625, 0.0125)$ | 5 | 24 | 75 |

Table 2: Hyperparameters used for ABS.

We use a DroQ critic with the original hyperparameters (Hiraoka et al., 2021) which consists of two critics with layers of size $(256, 256)$ with ReLU activations, and using LayerNormalization and Dropout. For all our experiments, we use an ensemble of 5 critics. We leverage the fact that the discounted policy returns are normalized and apply a *tanh* layer to bind the predictions of every critic between $[-1, 1]$. We use 32 parallel optimizers to optimize both the parameters of the Gaussian Process and the acquisition function. We follow Müller et al. (2021) and only train our GPs on the last $N_{max}$ data points collected in the Bayesian Optimization loop. In all our experiments, we keep track of the data generated by the last 3 outer loops of Algorithm 2, thus $N_{max} = 3 \times (1 + N_{acq})$ where $N_{acq}$ is the number of acquisition points sample at one outer loop. We also rollout the **central** point $\theta$ $N_c$ times in order to have a better advantage mean function estimate and to also alleviate the need for signal prior as described below.

The selection of hyperpriors for the signal variance $\sigma_f$ and the noise variance $\sigma_n$ requires domain knowledge about the task at hand and adds other hyperparameters to the BO problem. In the particular case of local BO, when we are at a fixed neighborhood it is fair to say that the observation noise is roughly similar among all points. The same logic applies to the signal variance. With this intuition in mind, we develop a heuristic to dynamically fix the hyperpriors for $\sigma_f, \sigma_n$. Using the fact that we roll out the **central** point $N_c$ times, we use a noise estimate $\widehat{\sigma}_n^2 = Var(\widehat{J}_1(\pi_\theta), \ldots, \widehat{J}_{N_c}(\pi_\theta))$ where $\widehat{J}_i$ is the noisy policy return estimate from the i-th rollout policy $\pi_\theta$. We then fix the hyperprior to be $\mathcal{U}[\frac{1}{3} \times \widehat{\sigma}_n, 3 \times \widehat{\sigma}_n]$. For the signal variance, we adopt a similar scheme by taking $\widehat{\sigma}_f^2 = Var(\{\hat{m}(x) - \hat{J}(\pi_x)\}_{x \in X})$ and use as a signal variance prior $\mathcal{U}[\frac{1}{3} \times \widehat{\sigma}_f, 3 \times \widehat{\sigma}_f]$. In the case of the constant mean used in MPD, the variance corresponds to the observation variance. MPD (Nguyen et al., 2022) has the merit of being less learning rate dependent than our method, being able to use a small learning rate and perform multiple descent steps. ABS cannot perform the same scheme as our advantage mean function depends on the **central** point $\theta$. In our implementation of MPD, we follow the code provided by the authors where we first normalize the gradient $g = \Sigma_\theta^- 1 \mu_\theta$ to have an L2 norm of 1 and use the same learning rate of $0.01$ for all environments. Hence, for MPD we perform grid-search only for the lengthscale hyperprior. For ABS, we perform a search over the learning rate and the lengthscale prior, and report the best-performing set of hyperparameters (Table 1). For all the results obtained with MPD and ABS in this manuscript, we used these best set of parameters (Table 2 and 3). For ARS, we use the original parameters reported in the paper (Mania et al., 2018).

| Task name | Learning rate | Lengthscale prior | $N_c$ | $N_{acq}$ | $N_{max}$ |
|---|---|---|---|---|---|
| InvertedPendulum | 0.01 | $\mathcal{U}(0.0025, 0.05)$ | 2 | 6 | 21 |
| Swimmer | 0.01 | $\mathcal{U}(0.0025, 0.05)$ | 3 | 12 | 39 |
| Hopper | 0.01 | $\mathcal{U}(0.0025, 0.025)$ | 3 | 16 | 51 |
| Halfcheetah | 0.01 | $\mathcal{U}(0.00125, 0.025)$ | 4 | 20 | 63 |
| Walker2d | 0.01 | $\mathcal{U}(0.000625, 0.0125)$ | 4 | 20 | 63 |
| Ant | 0.01 | $\mathcal{U}(0.000625, 0.0125)$ | 5 | 24 | 75 |

Table 3: Hyperparameters used for MPD.

# B  EXPERIMENTAL SECTION

## B.1  UNDISCOUNTED RETURNS

Here, we plot the evolution of the maximum undiscounted policy return as a function of the number of episodes. ABS, which can be seen as MPD augmented with the advantage mean function and the adaptive aggregation scheme consistently outperfoms MPD that uses the constant mean function. Our algorithm outperforms substancially the ARS baseline for the Walker2d and Ant tasks.

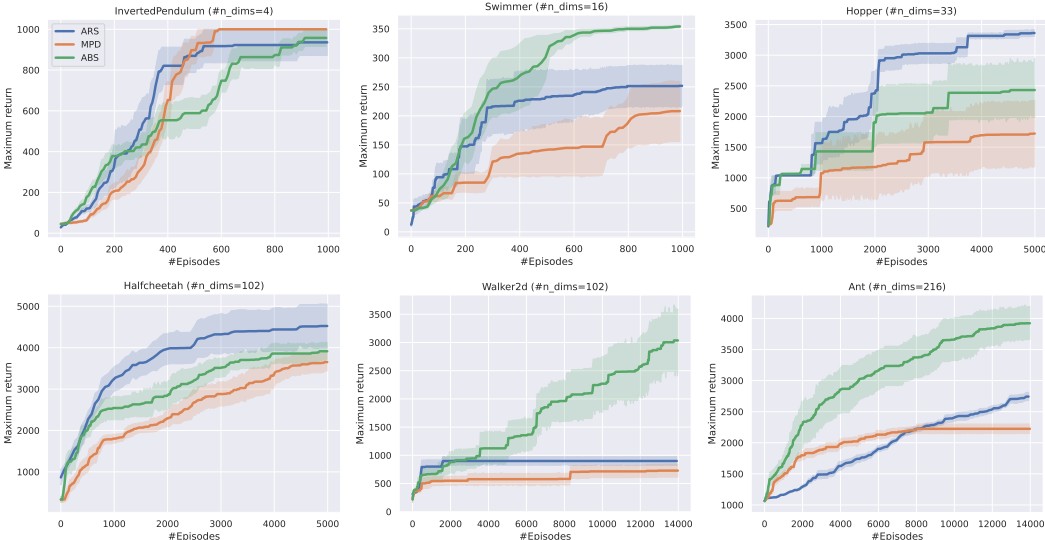

Figure 5: Evolution of the maximum undiscounted policy return on the MuJoCo-v4 tasks. We use 5 random seeds for every algorithm.

## B.2  VALIDATION AND TEST SCORES

Here, we plot the histogram of the distribution of the validation score $\widehat{R2}$ and test score $\widetilde{R2}$ used in Figure 2 for better visualization. We observe that the **validation** metric $\widetilde{R}^2$ remains consistently positive with the best critic explaining on average approximately $50\%$ of the variance observed data. The **test** metric $\widehat{R}^2$ manages also to be positive quite frequently, while being sometimes negative

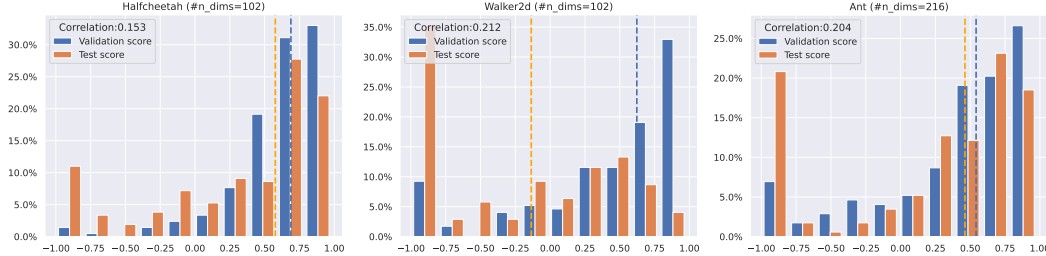

Figure 6: Histogram and correlation of the validation and test scores for MuJoCo-v4 tasks for a run of ABS. The dashed vertical lines represent the medians of their respective distributions. We clip the scores to [-1,1] for easier visualization.

## C PROOFS

### C.1 PROOF OF THE POSTERIOR OF THE GRADIENT

**Corollary C.0.1.** *Under the advantage mean function, the mean of the gradient posterior at $\theta$ is*

$$\mu_\theta = \mathbb{E}_{s\sim\widehat{d}^{\pi_\theta}}\left[\nabla_a\widehat{Q}_\phi^{\pi_\theta}(s,a)|_{a=\pi_\theta(s)}\nabla_\theta\pi_\theta(s)\right] + \nabla_\theta K(\theta,X)\,K(X,X)^{-1}(\widehat{J}(X)-\widehat{m}_\phi(X)). \quad (13)$$

*Proof.*

$$
\begin{aligned}
\nabla_\theta m_\phi(\theta) &= \frac{\partial m_\phi(x)}{\partial x}\bigg|_{x=\theta}\\
&\underset{(i)}{=} \frac{\partial}{\partial x}\left[J(\pi_\theta) + \mathbb{E}_{s\sim d^{\pi_\theta}}\left[A^{\pi_\theta}(s,\pi_x(s))\right]\right]\bigg|_{x=\theta}\\
&= \mathbb{E}_{s\sim d^{\pi_\theta}}\left[\frac{\partial}{\partial x}Q^{\pi_\theta}(s,\pi_x(s))\bigg|_{x=\theta} - \frac{\partial}{\partial x}Q^{\pi_\theta}(s,\pi_\theta(s))\bigg|_{x=\theta}\right]\\
&\underset{(ii)}{=} \mathbb{E}_{s\sim d^{\pi_\theta}}\left[\frac{\partial}{\partial x}Q^{\pi_\theta}(s,\pi_x(s))\bigg|_{x=\theta} - \frac{\partial}{\partial x}Q^{\pi_\theta}(s,\pi_\theta(s))\bigg|_{x=\theta}\right]\\
&= \mathbb{E}_{s\sim d^{\pi_\theta}}\left[\frac{\partial}{\partial x}Q^{\pi_\theta}(s,\pi_x(s))\bigg|_{x=\theta}\right]\\
&= \mathbb{E}_{s\sim d^{\pi_\theta}}\left[\frac{\partial}{\partial a}Q^{\pi_\theta}(s,a)\bigg|_{a=\pi_\theta(s)} \times \frac{\partial\pi_x(s)}{\partial x}\bigg|_{x=\theta}\right]
\end{aligned}
$$

Step (i) is due to independence of $J(\pi_\theta)$ and $x$. Step (ii) is true because $Q^{\pi_\theta}(\pi_\theta(s),s)$ is independent of $x$. We get the final result by injecting this equality into Equation 2. $\qquad\square$

### C.2 LIPSCHITZ MDPS INDUCE DISTRIBUTIONALLY LIPSCHITZ MDPS

**Assumption C.1** (Lipschitz MDP (Pirotta et al., 2015)). *An MDP with a policy class $\Pi$ is $(L_r, L_p, L_\pi)$-Lipschitz if for all $s, s' \in \mathcal{S}$ and $a, a' \in \mathcal{A}$:*

- ***Lipschitzness of reward:*** $|r(s,a) - r(s',a')| \le L_r(d_\mathcal{S}(s,s') + d_\mathcal{A}(a,a'))$

- ***Lipschitzness of transition:*** $W(p(.|s,a), p(.|s',a')) \le L_p(d_\mathcal{S}(s,s') + d_\mathcal{A}(a,a'))$

- ***Lipschitzness of policy:*** $W(\pi(s_1), \pi(s_2)) \le L_\pi d_S(s_1, s_2)$ for all $\pi \in \Pi$.

**Assumption C.2** (Distributionally Lipschitz MDP (Saleh et al., 2022)). *For an MDP with a policy class $\Pi$ is $(L_1, L_2)$-Lipschitz if for all $\pi_1, \pi_2 \in \Pi$ and $\mu_1, \mu_2 \in \mathcal{P}(\mathcal{S})$, the following holds true.*

- ***Lipschitzness w.r.t. policy:*** $W(P(\mu,\pi_1), P(\mu,\pi_2)) \le L_1 \sup_{s\in\mathcal{S}} W(\pi_1(s), \pi_2(s))$

- ***Lipschitzness w.r.t. state distribution:*** $W(P(\mu_1,\pi), P(\mu_2,\pi)) \le L_2 W(\mu_1, \mu_2)$

*Where $P$ denotes the generalization of the transition dynamics to take as input probability distributions of actions or states.*

**Assumption C.3.** *The state space $\mathcal{S} \in \mathbf{R}^d$ and is bounded.*

**Theorem C.4.** *Lipschitz MDPs (Assumptions C.1) induces Distributionally Lipschitz MDPs (Assumptions C.2) with Lipschitz rewards such that $L_1 = L_\pi$ and $L_2 = L_p(1 + L_\pi)$.*

*Proof.* Now, we prove that Assumptions C.1 (without Lipschitzness of reward) implies Assumptions C.2.

**Part 1: Lipschitzness w.r.t. policy.**

$$
\begin{aligned}
W(P(\mu, \pi_1), P(\mu, \pi_2)) &= W\left(\int_{\mathcal{S}} p(.|s, \pi_1(s))d\mu(s), \int_{\mathcal{S}} p(.|s, \pi_2(s))d\mu(s)\right) \\
&\underset{(a)}{\leq} \int W(a(s), b(s))d\mu(s) \\
&\leq \sup_{s \in \mathcal{S}} W(a(s), b(s)) \\
&\leq L_\pi \sup_{s \in \mathcal{S}} W(\pi_1(s), \pi_2(s)) \\
&= L_\pi W(\pi_1, \pi_2)
\end{aligned}
$$

Step (a) is due to Theorem 4.8 (Villani et al., 2009).

This implies that $L_1 = L_\pi$.

**Part 2: Lipschitzness w.r.t. state distribution.**

$$
\begin{aligned}
W_1(P(\mu_1, \pi), P(\mu_2, \pi)) &= W\left(\int_{\mathcal{S}} \underbrace{p(.|s, \pi(s))}_{\lambda(.|s)} d\mu_1(s), \int_{\mathcal{S}} p(.|s, \pi(s))d\mu_2(s)\right) \\
&= W\left(\int_{\mathcal{S}} \lambda(.|s)d\mu_1(s), \int_{\mathcal{S}} \lambda(.|s)d\mu_2(s)\right) \\
&\underset{(b)}{=} \sup_{\|f\|_L \leq 1} \left(\int_{\mathcal{S}} f(s')\left[\int_{\mathcal{S}} \lambda(s'|s)\mu_1(s)ds\right] ds' \right. \\
&\qquad\qquad\qquad \left. - \int_{\mathcal{S}} f(s')\left[\int_{\mathcal{S}} \lambda(s'|s)\mu_2(s)ds\right] ds'\right) \\
&= \sup_{\|f\|_L \leq 1} \int_{\mathcal{S}} f(s')\left[\int_{\mathcal{S}} (\mu_2(s) - \mu_1(s))\lambda(s'|s)ds\right] ds' \\
&\underset{(c)}{=} \sup_{\|f\|_L \leq 1} \int_{\mathcal{S}} (\mu_2(s) - \mu_1(s))\left[\underbrace{\int_{\mathcal{S}} f(s')\lambda(s'|s)ds'}_{g(s)}\right] ds \\
&\underset{(d)}{\leq} L_p(1 + L_\pi) \sup_{\|a\|_L \leq 1} \int_{\mathcal{S}} (\mu_2(s) - \mu_1(s))a(s)ds \\
&\leq L_p(1 + L_\pi)W(\mu_2, \mu_1).
\end{aligned}
$$

Step (b) is due to Remark 6.5 in Villani et al. (2009). Step (c) is obtained by applying Fubini-Tonelli theorem, as $\mathcal{S}$ is bounded. Step (d) holds true as $g$ is $L_p(1 + L_\pi)$-Lipschitz (Lemma C.5) and $f$ is maximum 1-Lipschitz.

Thus, we conclude that $L_2 = L_p(1 + L_\pi)$.

$\square$

**Lemma C.5.** *If we define $g(s) \triangleq \int_{\mathcal{S}} f(s')\lambda(s'|s)ds'$, for a Lipschitz MDP (Assumption C.1) and $f$ s.t $\|f\|_L \leq 1$ , then $g$ is $L_p(1 + L_\pi)$-Lipschitz.*

*Proof.*

$$|g(s_1) - g(s_2)| = \left| \int_{\mathcal{S}} f(s')\lambda(s'|s_1)ds' - \int_{\mathcal{S}} f(s')\lambda(s'|s_2)ds' \right|$$

$$\underset{(e)}{\leq} \sup_{\|h\|_L \leq 1} \left| \int_{\mathcal{S}} h(s')\lambda(s'|s_1)ds' - \int_{\mathcal{S}} h(s')\lambda(s'|s_2)ds' \right|$$

$$\leq W(\lambda(.|s_1), \lambda(.|s_2))$$

$$= W(p(.|s_1, \pi(s_1)), p(.|s_2, \pi(s_2)))$$

$$\leq L_p(d_{\mathcal{S}}(s_1, s_2) + d_{\mathcal{A}}(\pi(s_1), \pi(s_2))$$

$$\leq L_p(d_{\mathcal{S}}(s_1, s_2) + L_{\pi}d_{\mathcal{S}}(s_1, s_2))$$

$$\leq L_p(1 + L_{\pi}) \times d_{\mathcal{S}}(s_1, s_2)$$

$$\leq L_p(1 + L_{\pi}) \times d_{\mathcal{S}}(s_1, s_2)$$

Step (e) holds true because Lipschitz constant of $f$ is less than or equal to 1, and we take supremum over all such functions. □

### C.3 BOUND ON THE RESIDUAL TERM

Now, we use Theorem C.4 to bound the residual for Lipschitz MDPs.

**Theorem C.6.** *For a Lipschitz MDP satisfying (Assumptions C.1) and $\gamma L_P(1 + L_{\pi} < 1$, we have that the residual term*

$$\langle d^{\pi_2} - d^{\pi_1}, A^{\pi_1}(s, \pi_2(s)) \rangle \leq C \sup_{s \in \mathcal{S}} \|\pi_2(s) - \pi_1(s)\|, \qquad (14)$$

*for $C \triangleq \frac{2\gamma L_{\pi} L_r (1 + L_{\pi})}{(1 - \gamma L_P(1 + L_{\pi}))^2}$.*

*Proof.* **Step 1:** First, we bound the residual using the Wasserstein distance between policies and $L_2$ distance between the gradient of $Q$ functions.

$$\langle d^{\pi_2} - d^{\pi_1}, A^{\pi_1}(s, \pi_2(s)) = \langle d^{\pi_2} - d^{\pi_1}, Q^{\pi_1}(s, \pi_2(s)) - Q^{\pi_1}(s, \pi_1(s)) \rangle$$

$$\leq W(d^{\pi_2}, d^{\pi_1}) \times \sup_s \|\nabla_s [Q^{\pi_1}(s, \pi_2(s)) - Q^{\pi_1}(s, \pi_1(s))]\| \quad (15)$$

We obtain the last inequality from the Kantorovich-Rubinstein inequality (Villani et al., 2009).

**Step 2:** Now, we bound the the change in gradient of Q-functions due to the change in policies.

$$\nabla_s [Q^{\pi_1}(s, \pi_2(s)) - Q^{\pi_1}(s, \pi_1(s))]$$

$$= \frac{\partial Q^{\pi_1}}{\partial S}\Big|_{s, \pi_2(s)} + \nabla_S \pi_2(s) \frac{\partial Q^{\pi_1}}{\partial a}\Big|_{s, \pi_2(s)} - \frac{\partial Q^{\pi_1}}{\partial S}\Big|_{s, \pi_1(s)} - \nabla_S \pi_1(s) \frac{\partial Q^{\pi_1}}{\partial a}\Big|_{s, \pi_1(s)}$$

$$= \underbrace{\frac{\partial Q^{\pi_1}}{\partial S}\Big|_{s, \pi_2(s)} - \frac{\partial Q^{\pi_1}}{\partial S}\Big|_{s, \pi_1(s)}}_{a} + \underbrace{\nabla_S \pi_2(s) \frac{\partial Q^{\pi_1}}{\partial a}\Big|_{s, \pi_2(s)} - \nabla_S \pi_1(s) \frac{\partial Q^{\pi_1}}{\partial a}\Big|_{s, \pi_1(s)}}_{b}$$

**Step 3: Bounding Term a.** Here, we leverage the property that if $f$ is $L$-Lipschitz, $\|\nabla f\| \leq L$.

$Q$ function is $L_Q$-Lipschitz for a Lipschitz MDP, such that $L_Q \triangleq \frac{L_r}{1 - \gamma L_P(1 + L_{\pi})}$ for $\gamma L_P(1 + L_{\pi}) < 1$ (Pirotta et al., 2015).

Thus, we conclude that $\left\| \frac{\partial Q^{\pi_1}}{\partial S}\Big|_{s,a} \right\| \leq L_Q$. This bounds the Term a by $2L_Q$.

**Step 4: Bounding Term b.**

$$\left\| \nabla_s \pi_2(s) \frac{\partial Q^{\pi_1}}{\partial a}\Big|_{s, \pi_2(s)} \right\| \leq \|\nabla_s \pi_2(s)\| \times \left\| \frac{\partial Q^{\pi_1}}{\partial a}\Big|_{s, \pi_2(s)} \right\| \leq L_Q L_{\pi}$$

The first inequality is a consequence of Hölder's inequality, while the second one is obtained from Lipschitzness of $Q$ and $\pi$. Thus, Term b is bounded by $2L_Q L_\pi$.

**Step 5: Putting it all together.** Combining Equation (15) with the bounds on Term a and Term b yields

$$
\begin{aligned}
\langle d^{\pi_2} - d^{\pi_1}, A^{\pi_1}(s, \pi_2(s)) \rangle &\leq W(d^{\pi_2}, d^{\pi_1}) \times 2\, L_Q(1\ +\ 2L_\pi) \\
&\underset{(a)}{\leq} \frac{\gamma L_\pi}{1 - \gamma L_p(1 + L_\pi)} \sup_{s \in \mathcal{S}} W(\pi_1(s), \pi_2(s))\ \times\ 2\, L_Q(1 + L_\pi) \\
&= \frac{2\gamma L_\pi L_r\,(1 + L_\pi)}{(1 - \gamma L_p(1 + L_\pi))^2} \sup_{s}\ \|\pi_2(s) - \pi_1(s)\|
\end{aligned}
$$

Step (a) is derived by an application of Theorem B.4 in (Saleh et al., 2022), when $\gamma L_p(1 + L_\pi) < 1$.

$\square$

