# OpenReview forum: "Augmented Bayesian Policy Search"
_ICLR.cc/2024/Conference — ICLR 2024 poster_

### Official Review · Reviewer_fyPW · 2023-10-31

**Soundness:** 4 excellent
**Presentation:** 4 excellent
**Contribution:** 3 good
**Rating:** 6
**Confidence:** 3

**Summary:**

The authors popose Augmented Bayesian policy Search (ABS) a Bayesian optimisation (BO) algorithm for reinforcement learning (RL). Typically, BO methods for RL estimate the objective function (dsicounted rewards) using Gaussian Processes (GP) with a constant mean function for the GP prior. In this work the authors propose a new mean function for the GP prior based on empirical estaimtes of the objective function and the advantages between the old policy and new policy. By using this advantage mean function the authors show that the gradient of the posterior takes the form of the deterministic policy gradient corrected by how closely the mean function fits the observed dataset. This approach is theoretically backed by a new bound derived by the aithors that bounds the residual term of a rearranging of the performance difference lemma for Lipschitz MDPs. Finally, the authors also develop an adaptive scheme for aggregating ensembles of Q function estimators based on fitness measure (R-squared) on the whole dataset.

**Strengths:**

(1) The paper is well motivated and a comprehensive introduction and related work section helps the reader grasp where this research sits in the literature.

(2) The contribution is novel and significant. Specifically, the new bound based on the performance difference lemma for Lipschitz MDPs is convincing and useful (although I haven't thoroughly checked the details on this). The results are also promising and demonstrate good sample complexity.

(3) The algorithm itself is well motivated and theoretically backed, and the evaluation is sufficient enough for me.

(4) To me the paper seems mathematically sound, presented well and well written.

I find this an interesting paper and I'm sure it would be of interest at ICLR, which is why I am recommending it for acceptance. Although since I am not an expert in BO for RL I only feel able to weakly accept this paper. Hopefully a more well read reviewer can see more of the merit of this paper.

**Weaknesses:**

(1) Quite a lot of prior knowledge is required to understand this paper. I am not immediately familarly with BO methods for RL, perhaps some more intuition for comparison to more familair RL methods may be helpful for the average RL practitioner.

**Questions:**

I am not familiar with Lipschitz MDPs, presumable the reward function is Lipschitz bounded, but what of the transition functions and policies? Is there a natural way of bounding these that makes sense?

What are the key difference between your approach and the baselines considered in this paper?

What are the reasons for only considering deterministic policies?

---

> ### Author Response · Authors · 2023-11-15
>
> > (1) Quite a lot of prior knowledge is required to understand this paper. I am not immediately familiar with BO methods for RL, perhaps some more intuition for comparison to more familiar RL methods may be helpful for the average RL practitioner.
>
> We agree with the reviewer, we chose the MPD algorithm which adds another layer on top of the usual BO algorithms for two reasons. First, it is the state-of-the-art method in BO. The second is that MPD is a First Order BO method which allows us to derive Corollary (3.3.1) which bridges the gap between BO and RL methods.
>
> Our work builds on MPD, which is a local Bayesian optimization method. This line of research is very recent and combines Bayesian optimization with gradient descent. Our work specifies and improves this method for RL (prior work was completely black-box and could be applied to any domain) by using a learned Q-function in the Gaussian process prior. At an intuitive level, our method can be seen as a policy gradient method [1], where Bayesian optimization techniques are used to select in a clever way the batch of deterministic policies that we rollout to collect transitions from the environment and hence explore.
>
> > (2) I am not familiar with Lipschitz MDPs, presumable the reward function is Lipschitz bounded, but what of the transition functions and policies? Is there a natural way of bounding these that makes sense?
>
> For a Lipschitz MDP, the transition dynamics is Lipschitz continuous w.r.t. Wasserstein distance. The same applies to the policies whose actions are smooth w.r.t to the state. This assumption has been used in the literature [2] to bound the residual term for deterministic policies. We refer the reviewer to Assumptions 3.1 and 3.2 for further details.
> Though our theoretical results rely on this assumption, we do not need to compute the Lipschitz constants for our algorithm, as demonstrated by our experimental results for MuJoCo.
>
> > (3) What are the key differences between your approach and the baselines considered in this paper?
>
> Our main baseline is the MPD algorithm. Our algorithm ABS can be seen as a variant where we improve the mean function of the Gaussian Process, usually taken to be a constant, to the advantage mean function that we develop.
>
> The second baseline ARS is a standard evolutionary strategy algorithm. We are different from ARS in that our algorithm will be sampling adaptively the "acquisition points" using the acquisition function, whereas ARS samples the acquisition points according to a fixed distribution. ARS also follows a different descent direction for the gradient.
>
> > (4) What are the reasons for only considering deterministic policies?
>
> Our analysis and methodology generalize to stochastic policies (except Equation (7)).
>
> We choose to benchmark our algorithm using linear policies for 3 reasons:
>
> 1) In our study, we focus on learning good policies while using exclusively deterministic policies for exploration. This approach is particularly beneficial when interacting with physical systems, where the implementation of deterministic policies is often more desirable.
>
> 2) RL with deterministic linear policies is a standard benchmark for the Bayesian optimization setting [3,4,5].
>
> 3) Deterministic policies allow for less noise in the policy evaluation compared to a stochastic policy and hence fewer rollouts are necessary to estimate the return of a policy $\pi$.
>
> [1] Sutton, Richard S., and Andrew G. Barto. Reinforcement learning: An introduction. MIT press, 2018.
>
> [2] Saleh, Ehsan, et al. "Truly Deterministic Policy Optimization." Advances in Neural Information Processing Systems 35 (2022): 8469-8482.
>
> [3] Müller, Sarah, Alexander von Rohr, and Sebastian Trimpe. "Local policy search with Bayesian optimization." Advances in Neural Information Processing Systems 34 (2021): 20708-20720.
>
> [4] Nguyen, Quan, et al. "Local Bayesian optimization via maximizing probability of descent." Advances in neural information processing systems 35 (2022): 13190-13202.
>
> [5] Eriksson, David, et al. "Scalable global optimization via local Bayesian optimization." Advances in neural information processing systems 32 (2019).

---

> > ### Comment · Reviewer_fyPW · 2023-11-16
> >
> > Thanks, for your answers - they have helped clarify a few things for me. I think this is an interesting paper, I will be keeping my score the same since I am not immediately familiar with the literature around BO for RL, so I can't attest to this papers contributions. Best of luck.

---

> > ### Comment · Reviewer_yvFW · 2023-11-19
> > **Lipschitz MDPs**
> >
> > The requirement for a Lipschitz MDP is presumably intended to avoid the problems described in (Wang et al. 2023).
> >
> > Wang et al., Fractal Landscapes in Policy Optimization, NeurIPS 2023.

---

> > > ### Author Response · Authors · 2023-11-23
> > >
> > > We thank the Reviewer for sharing this reference. It is indeed observed that stochastic policies tend to smooth the policy return landscape [1]. This characteristic can potentially mitigate the effects highlighted in the provided reference [2]. In contrast, deterministic policies are often faced with a more challenging task as they navigate a comparatively rougher landscape. In our work, we made the Lipschitz assumption in the theoretical part of the paper to introduce enough structure to be able to state meaningful theoretical results.
> > >
> > > Indeed, in the case of stochastic policies, there is always an overlap of probability density in the trajectory space for two different policies. This trajectory overlap allows bounding the difference in policy return between two policies, and it is a consequence of having overlapping probability distributions over actions. When the policies are deterministic, two policies might have no overlap at all in trajectory space even though they take similar actions. Without introducing any structure in the MDP, it would be hard to say anything in the worst case. Hence, we need to at least ensure that if two policies take close enough actions, they will visit close enough states.
> > >
> > > However, we would like to emphasize that this assumption is only used in the theoretical analysis and not in our algorithm.
> > >
> > > In our experimental section, we do not need the information on whether the MDP is Lipschitz nor make use of any Lipschitz constant. Moreover, the locomotion tasks are known to have non-Lipschitz dynamics [3].
> > >
> > > [1] Ahmed, Zafarali, et al. "Understanding the impact of entropy on policy optimization." International conference on machine learning. PMLR, 2019.
> > >
> > > [2] Wang et al., Fractal Landscapes in Policy Optimization, NeurIPS 2023.
> > >
> > > [3] Weng, Bowen, et al. "On safety testing, validation, and characterization with scenario-sampling: A case study of legged robots." 2022 IEEE/RSJ International Conference on Intelligent Robots and Systems (IROS). IEEE, 2022.

---

> > > > ### Comment · Reviewer_yvFW · 2023-11-23
> > > >
> > > > A very interesting discussion. Many thanks for the further references. I have the feeling (I can't put it into words properly right now) that these problems also increase the significance of the present work.

---

### Official Review · Reviewer_m5Gu · 2023-11-01

**Soundness:** 3 good
**Presentation:** 3 good
**Contribution:** 3 good
**Rating:** 6
**Confidence:** 4

**Summary:**

This paper presents an elegant BO framework that combines policy gradient and policy evaluation information.  The paper leverages a new construct, the advantage mean function, to improve the typical mean function used in Gaussian processes; this mean function leverages the nature of RL to improve over a naive mean function, and allows the GP to focus its modeling power on second-order residuals.  By jointly modeling the mean and gradient exploration can be improved.

Overall, I really liked this paper. I felt like it presented a strong combination of ideas that are novel, creative, and likely to have impact in the field.

I only gave it a 6 because I have several questions about the method that I could not resolve by reading the paper.

**Strengths:**

+ An elegant combination of multiple ideas
+ The advantage mean function is a good idea
+ Love the idea of jointly modeling the value and gradient with a single GP
+ Clearly written (although reader needs to be an expert), well-referenced, etc.

**Weaknesses:**

- The paper is somewhat dense and difficult to parse.
- Empirical results are fine, but could be stronger.

**Questions:**

I am a little bit worried about the assumption of a linear policy; it was unclear to me if this was for exposition, or a constraint on the algorithm. Could you clarify?

I see that you assume that you have access to J, d, and Q, but not more granular information about observed s,a,r sequences. Why is that? Is that just an assumption that could be relaxed, and if so, would the more granular information be useful in constructing better GP posteriors?

You combine the Q-experts using a softmax of exp(R^2) terms. This seems somewhat ad hoc. Is there a principled reason for this choice?

Resetting the least-performing critic also seems like an ad hoc choice. Is there a principled reason for it?

---

> ### Author Response · Authors · 2023-11-15
>
> > (1) The paper is somewhat dense and difficult to parse.
>
> We have updated Section 4 for simpler exposition. Please let us know if the paper is clearer and if there are any other parts of the paper that you would like to be clarified.
>
> > (2) Empirical results are fine but could be stronger.
>
> We would like to point out that our algorithm consistently outperforms the MPD baseline, which, to the best of our knowledge, is the state-of-the-art algorithm in Bayesian Optimization (BO) for Reinforcement Learning (RL). Notably, our mean function can improve all potential future BO methods when applied to continuous RL problems.
>
> Also, we would like to point out that despite performing slightly worse than ARS in Hopper and Halfcheetah, the gap between our method and ARS is never large in ARS' favor. In contrast, on the higher dimensional tasks that are Walker2d and Ant, our method largely outperforms ARS.
>
> > (3) I am a little bit worried about the assumption of a linear policy; it was unclear to me if this was for exposition, or a constraint on the algorithm. Could you clarify?
>
> Our theoretical analysis and methodology are not dependent on the linear policy assumption. Assumption 3.2 requires the policy to be Lipschitz and this includes the richer class of policies including the linear ones (e.g. a ReLU neural network can be Lipschitz).
>
> We choose to benchmark our algorithm using linear policies for two reasons:
>
> 1) RL with deterministic linear policies is a standard benchmark for the Bayesian optimization setting [1,2,3].
>
> 2) For computational considerations, BO cannot scale to the thousands of dimensions required for a neural network actor. At least, not with the standard algorithms.
>
> > 4. I see that you assume that you have access to J, d, and Q, but not more granular information about observed s,a,r,s' sequences. Why is that? Is that just an assumption that could be relaxed, and if so, would the more granular information be useful in constructing better GP posteriors?
>
> We train the Q-function in the classical way on the available transition data $\langle s,a,r,s' \rangle$ collected from the rolled-out trajectories. Thank you for pointing this out. We make this point clearer in the revision of Algorithm (1).
>
> > (5) You combine the Q-experts using a softmax of $\exp(R^2)$ terms. This seems somewhat ad hoc. Is there a principled reason for this choice?
>
> Our aggregation scheme for critics is an adoption of the Follow The Regularized Leader (FTRL) algorithm to average our critics using softmax weighting and $R^2$ as the goodness-of-fit function [4]. We clarify this point in the revised draft. We would also like to point out that the ablation study benchmarks a variant of our algorithm with a uniform weighting of the Q-experts (label Ensemble+R in Figure 4).
>
> > (6) Resetting the least-performing critic also seems like an ad-hoc choice. Is there a principled reason for it?
>
> In practice, we noticed some high variance between different runs when using a single critic. This was due to the critic neural network losing plasticity as discussed in the primacy bias paper. Resetting critics also helps decorrelating their predictions, which is useful for ensemble-based methods. In the ablation study, we compare our algorithm to an ablated version that removes the resetting of critics (label Softmax in Figure 4). We hope this makes a sufficiently convincing argument about the practical use of this choice.
>
> [1] Müller, Sarah, Alexander von Rohr, and Sebastian Trimpe. "Local policy search with Bayesian optimization." Advances in Neural Information Processing Systems 34 (2021): 20708-20720.
>
> [2] Nguyen, Quan, et al. "Local Bayesian optimization via maximizing probability of descent." Advances in neural information processing systems 35 (2022): 13190-13202.
>
> [3] Eriksson, David, et al. "Scalable global optimization via local Bayesian optimization." Advances in neural information processing systems 32 (2019).
>
> [4] Cesa-Bianchi, Nicolo, and Gábor Lugosi. Prediction, learning, and games. Cambridge university press, 2006.

---

### Official Review · Reviewer_ysm5 · 2023-11-02

**Soundness:** 3 good
**Presentation:** 3 good
**Contribution:** 3 good
**Rating:** 8
**Confidence:** 4

**Summary:**

The authors present a new method for policy search, combining local Bayesian optimization with policy search. In essence, they describe novel mean function for a Gaussian process that is derived from the performance difference lemma.

Overall, the algorithm performs reasonably well and appears to outperform existing methods on some tasks

**Strengths:**

- the paper is well written, contributions are highlighted, clear experimental questions
- while not easy to follow (one needs to have expertise in many different areas: model-free RL, (local) Bayesian Optimization, Gaussian processes) the author explain their reasoning well
- the experiments  are carried out rigourosly, including repetitions, definition of research questions, etc
- the finding is novel: utilizing the performance difference lemma to derive a more informed mean function to have a better estimate of the performance of neighboring policies is clever (in some ways an informed Taylor approximation)

**Weaknesses:**

1. minor: The plots can be improved, figure 2-4  is not very pretty (too high linewidth, scrollbar on the right, grid overrides plot)
2. Performance is only marginally better/worse than existing methods

3. This is not a weakness/critique directly but it relates to point 2. Also this point will be a bit opinionated.

I do not believe that the "classical RL setup" of:  exploration towards exploitation(i.e. gradually improving/figuring out the task) is where this method is applied best. Essentially you use the BO + the local mean function interpolation to iteratively improve the policy estimation.    That is why, comparing to "less rigorous/Bayesian" approaches the improvements are not very striking

However, I believe a stronger case can be made if you would look at the method from a risk-sensitive perspective. Your mean function will give you an informed estimate around the known evaluation. The more you move away from that the more the epistemic uncertainty of the probabilitc model will set in. This would enable you to essentially define a "safe zone" around your current estimate.  And your method would have a better informed safe zone compared to other approaches because you leverage the information better using the advantage mean function. Therefore I would expect that your method would still maintain the advantages of a risk-sensitive method (i.e. by monotonic improvement over time) but be more "quick"/greedy compared to traditional risk-sensitive approaches.

4. Related,  conceptually speaking   you make your mean function more informed utilizing the performance difference lemma. This, essentially is a form of interpolation from existing points (i.e. estimating the return of a policy \phi, given knowledge of the return of policy \theta.).  Such knowledge/modelling naturally would go to the kernel function, instead.  The result of your current approach, infusing it in the mean funciton instead,  is that while you mean function is more informed, your covariance function is not.  Therefore, your covariance function will only be a function of the standard kernel function and X.  This means, your estimate of predictive epistemic uncertainty does not take into account the insight "using the knowledge of \theta I can make a more informed guess about the policy \phi". So in my opinion you are still underutilizing the insight described in this paper.

**Questions:**

Did you see that compared to the baseline your approach gave a more stable/monotonic improvement over time?

---

> ### Author Response · Authors · 2023-11-15
>
> > (1) Minor: The plots can be improved, figure 2-4 is not very pretty (too high linewidth, scrollbar on the right, grid overrides plot)
>
> Thank you for pointing this out. We have updated the figures now. Let us know if the updated figures are well-suited.
>
> > (2) Performance is only marginally better/worse than existing methods
>
> We point out that our algorithm consistently outperforms the MPD baseline, which, to the best of our knowledge, is the state-of-the-art algorithm in BO for RL. By doing so, we argue that our mean function can improve all potential future BO methods when applied for RL which would be more the takeaway message. Also, we would like to point out that despite performing slightly worse than ARS in Hopper and Halfcheetah, the gap between our method and ARS is never large in ARS' favor. In contrast, on the higher dimensional tasks that are Walker2d and Ant, our method largely outperforms ARS.
>
> > (3) I believe a stronger case can be made if you would look at the method from a risk-sensitive perspective.
>
> Thank you for raising this interesting point. We believe this is a promising direction that we aim to explore in future works. We refer the Reviewer to our response to point 5.B for additional discussion on the risk-sensitive perspective.
>
> > (4) Your covariance function will only be a function of the standard kernel function.
>
> Thank you for raising another interesting point. As we mention in our conclusion, the kernel function is a direction we plan to explore for future works.
>
> > (5) Did you see that compared to the baseline your approach gave a more stable/monotonic improvement over time?
>
> When observing individual seeds, we did not notice more monotonic behavior of our algorithm compared to MPD and ARS for both the acquisition phase and when comparing the return from one central point to the other. Our explanation for this is threefold:
>
> - A) The gradient of ARS can be seen as that of a robust optimization objective which justifies a fairly good monotonic increase. The same applies to MPD, which follows the maximum descent direction and hence demonstrates quite a consistent increase.
>
> - B) Another phenomenon is that the acquisition function is symmetric w.r.t to the posterior of the gradient $\mu_\theta$. Knowing that the direction $-d$ is bad is equivalent to knowing that direction $+d$ is good. Breaking the symmetry of the acquisition function to favor the good direction $+d$ would be an interesting idea for a risk-sensitive BO algorithm.
>
> - C) Finally, the monotonic behavior is very linked to the chosen learning rate. Making it harder to compare the behavior of the algorithms.

---

### Official Review · Reviewer_yvFW · 2023-11-03

**Soundness:** 3 good
**Presentation:** 2 fair
**Contribution:** 2 fair
**Rating:** 8
**Confidence:** 4

**Summary:**

The paper deals with model-free, bootstrapping-free online RL. It investigates Bayesian optimization for policy search. The new method is compared with two similar methods on six deterministic benchmark MDPs and shows similar results.

**Strengths:**

* The approach is interesting, I think bootstrapping-free approaches should be given more attention.

**Weaknesses:**

* There is no comparison with one of the frequently used model-free RL methods with bootstrapping.
* The presentation does not classify the approach clearly enough in the very broad field of RL algorithms.
* The results do not show a clear superiority over the two comparative methods. It is not made sufficiently clear why this approach should nevertheless receive attention.
* The method is only tested on deterministic MDPs without mentioning this limitation.

Further comments:

* The term "deterministic exploration" requires an explanation.

* Regarding "deterministic policy gradient, which changes its actions to maximize Qπθ (Silver et al., 2014)", I would like to note that this principle was already introduced in 2007 in [1].

* "max" should not be italicized, see Algorithm 1.

* The sentence "Evolution of the validation and test scores a representative seed on the MuJoCo task" is incomprehensible.

* The claim "Our learned Q-function can reconstruct the real Q-function" is, in my opinion, not sufficiently substantiated by Figure 2.

* In my opinion, "In the direct policy search literature, an alternative to BO is the use of Evolutionary Strategies" lacks the mention of the use of swarm-based methods such as PSO in [2] and [3].


* There are several unintentional lower case letters in the references: bayesian, gaussian, lipschitz, markov

[1] Schneegaß et al., Improving optimality of neural rewards regression for data-efficient batch near-optimal policy identification, 2007\
[2] Hein et al., Reinforcement learning with particle swarm optimization policy (PSO-P) in continuous state and action spaces, 2016\
[3] Hein, Interpretable Reinforcement Learning Policies by Evolutionary Computation, 2019

**Questions:**

* Why was a comparison with bootstrapping-based methods such as NFQ [4], DQN [5], SAC [6], etc. omitted?
* Why was a comparison with a bootstrapping-free, model-based method such as PILCO [7] omitted?

[4] Riedmiller, Neural Fitted Q Iteration – First Experiences with a Data Efficient Neural Reinforcement Learning Method, 2005\
[5] Mnih et al., Playing Atari with Deep Reinforcement Learning, 2013\
[6]  Haarnoja et al.,  Soft Actor-Critic: Off-Policy Maximum Entropy Deep Reinforcement Learning with a Stochastic Actor, 2018\
[7] Deisenroth and Rasmussen, PILCO: A Model-Based and Data-Efficient Approach to Policy Search, 2011

---

> ### Author Response · Authors · 2023-11-15
>
> > (1) Why was a comparison with bootstrapping-based methods such as NFQ [4], DQN [5], SAC [6], etc. omitted?
>
> We kindly point out that the aim of our paper is to construct an effective algorithm for RL that explores using only deterministic policies. It is an important property of policy search methods, such as Bayesian optimization or random search, when applied to physical systems as discussed in the introduction of our paper. As also cited in our introduction, we consider the problem of continuous MDPs. Hence, methods like NFQ and DQN do not apply.
>
> As for SAC and other bootstrapping-based methods, they do not fit into our problem of performing exploration using only deterministic policies, as mentioned.
>
> > (2) The presentation does not classify the approach clearly enough in the very broad field of RL algorithms.
>
> We improved the introduction section to better stress the use case of our algorithm. In broad terms, in this paper, we are interested in policy search algorithms that only execute deterministic policies on the controlled system. This includes two main classes: random search algorithms and Bayesian optimization. MPD, which our work builds on, is a Bayesian optimization algorithm and our main competitor in the experiments. One important contribution of our paper is that neural Q-functions can be used to improve the probabilistic modeling of Bayesian optimization. To support this statement, we have shown that our algorithm outperforms MPD on all the tasks.
>
> > (3) The results do not show a clear superiority over the two comparative methods.
>
> We would like to point out that our algorithm consistently outperforms the MPD baseline, which, to the best of our knowledge, is the state-of-the-art algorithm in BO for RL. Because our improvements are only on the probabilistic modeling of the policy return but not other components of BO, such as the acquisition function, we think that our proposed mean function can potentially improve many future BO methods when applied for RL, which we consider as our main contribution.
>
> Also, we want to point out, that albeit performing slightly worse than ARS in Hopper and Halfcheetah, but we outperform all the competing algorithms by a large margin for Walker2d and Ant.
>
> > (4) The method is only tested on deterministic MDPs without mentioning this limitation.
>
> We raise the point that the majority of BO works applied in RL for continuous MDPs use the MuJoCO benchmark. Nevertheless, we want to remark that our theoretical results apply for both deterministic and stochastic MDPs.
>
> The fundamental challenges of learning in stochastic and deterministic MDPs with deterministic policies are similar in nature. For stochastic MDPs, due to much noisier evaluations, one can use more samples to construct the empirical estimates (e.g., policy return and discounted state occupation measure).
>
> > (5) The term "deterministic exploration" requires an explanation.
>
> We fixed this in the introduction. Thank you for pointing this out.
>
> > (6) Regarding "deterministic policy gradient ...", I would like to note that this principle was already introduced in 2007 in [1].
>
> We want to kindly point out that the aforementioned paper did not formally justify the use of the policy gradient for deterministic policies. This was initially introduced in the "Deterministic policy gradient" paper (Silver et al., 2014).
>
> > (7) "max" should not be italicized, see Algorithm 1.
>
> We fixed that. Thank you for your comment.
>
> > (8) The sentence "Evolution of the validation and test scores a representative seed on the MuJoCo task" is incomprehensible.
>
> It should have read "Evolution of the validation and test scores of a representative seed on the MuJoCo task". We have fixed that. Thank you for your feedback.
>
> > (9) The claim "Our learned Q-function can reconstruct the real Q-function" is, in my opinion, not sufficiently substantiated by Figure 2.
>
> We agree with the reviewer that this statement is inaccurate. We decided to remove this claim from the revised version.
>
> > (10) In my opinion, ... lacks the mention of the use of swarm-based methods such as PSO in [2] and [3].
>
> Thank you. We have added these citations in the related works section of the revised paper.
>
> > (11)  unintentional lower case letters
>
> Thank you for pointing this out. We have fixed this.
>
> > (12) Why was a comparison with a bootstrapping-free, model-based method such as PILCO  omitted?
>
> To the best of our knowledge, there has been no prior work that successfully applied PILCO on MuJoCO locomotion tasks, and prior work also noted more generally the lack of success of model-based methods using Gaussian processes for locomotion with frictional contacts [1] such as the MuJoCO tasks we consider.
>
> [1]: Nagabandi, Anusha, et al. "Neural network dynamics for model-based deep reinforcement learning with model-free fine-tuning." ICRA, 2018.
>
> [2]: Deisenroth, Marc Peter, et al. "Toward fast policy search for learning legged locomotion."  IROS, 2012.

---

> > ### Comment · Reviewer_yvFW · 2023-11-17
> > **lower case letters**
> >
> > >> (11) unintentional lower case letters
> >
> > > Thank you for pointing this out. We have fixed this.
> >
> > This may sound pedantic, and I will adress some more important topics in a seperate comment, but still I want to point out here, that the wrong lower case letters are still in the PDF. They might not be there in the bib file. You need to use braces, like {Bayesian} to fix this.

---

> > > ### Author Response · Authors · 2023-11-17
> > >
> > > Thank you. We have uploaded a corrected revised version, where we hope to have fixed all remaining lower cases.

---

> > ### Comment · Reviewer_yvFW · 2023-11-17
> > **Comparison**
> >
> > > As for SAC and other bootstrapping-based methods, they do not fit into our problem of performing exploration using only deterministic policies, as mentioned.
> >
> > Nevertheless, I would like to see a comparison of the performance, or even better the performance as a function of the number of episodes, as in Fig. 3 with a standard technique such as SAC. Even if the comparison would be unfair in the sense that SAC uses stochastic policies, the comparison would make it possible to better contextualize the performance of the BO methods.
> > I think the paper is good enough to be published even without this comparison. However, my feeling is that it would gain massively from such a comparison.

---

> > ### Comment · Reviewer_yvFW · 2023-11-17
> > **adjusted assessment**
> >
> > The changes have noticeably improved the paper in my opinion. Together with the explanations, I am now in favor of accepting the paper.

---

### Official Review · Reviewer_gD9d · 2023-11-04

**Soundness:** 2 fair
**Presentation:** 3 good
**Contribution:** 2 fair
**Rating:** 6
**Confidence:** 4

**Summary:**

This paper presents a method for policy search/actor critic based on local Bayesian optimization. The main contribution is in the use of a mean function that captures the advantage (critic) of the model  that relates a set of parameters theta with the queried set of parameters x.

**Strengths:**

The idea of using the recent local Bayesian optimization methods for efficient policy search for high dimensional problems is quite interesting. Furthermore, it is well integrated with the performance difference lemma to enable updates on the information of theta with queries at x. I can see a lot of potential of this work in the community.

**Weaknesses:**

There are some limitations that can reduce the applicability of this work and make it difficult to understand. For example, it is never explained the type of policy being used. The derivation is based on nonparametric (tabular) policies, but those type of policies are too limited for the MuJoCo experiments. Even the linear policy seems to be very limited.
The idea of incorporating information from x at theta is interesting, but that still requires to sample rollouts from the policy pi_x, which seems counterintutive in a BO setting where sample efficiency is of paramount importance. It seems that the philosophy behind the mean function is similar to having a GP with inducing points. The other advantage of using pi_x seems exploration, but that can also be achieved with importance sampling.
Given the nature of the application (sample efficient PS), the authors should include in the comparison both novel AC methods (SAC, TQC…), model based approaches which are also sample efficient (MBPO…). For example, according to the MBPO paper [A], MBPO is able to achieve 6000 reward in a similar number of steps than ABS gets 4000.

[A] Janner M, Fu J, Zhang M, Levine S. When to trust your model: Model-based policy optimization. Advances in neural information processing systems. 2019;32.

**Questions:**

-In the paragraph before eq 8, it says “Proposition 3.3”. I think you mean “Theorem 3.3”
-Why all the policies in Figs 3 and 4 are normalized? What do you mean by discounted policy returns? It seems that the undiscounted returns like the ones presented in the appendix are more standard in the literature and allows comparisons with other works. Also, the standard approach is to use steps for the horizontal axis, instead of episodes.
-The R2 analysis is unclear. The fact that for many iterations the correlation is negative and it is mostly below 0.5 means that the fitness might not be accurate. Furthermore, the R2 is a fitness measurement for linear regression, which does not seem to be the case for the Q-function.
-Policy search with Bayesian optimization has been extensively studied since 2007 [B] and there are many works missing. For example, [C] presented a PS method with BO using local models as well, while [D] also combines parametric and non-parametric (GP) models.

[B] Martinez-Cantin R, de Freitas N, Doucet A, Castellanos JA. Active policy learning for robot planning and exploration under uncertainty. In Robotics: Science and systems 2007 Jun (Vol. 3, pp. 321-328).

[C] Martinez-Cantin R. Funneled Bayesian optimization for design, tuning and control of autonomous systems. IEEE transactions on cybernetics. 2018 Feb 27;49(4):1489-500.

[D] Englert P, Toussaint M. Combined Optimization and Reinforcement Learning for Manipulation Skills. In Robotics: Science and systems 2016 Jun (Vol. 2016).

----
After discussion comment: After reading the authors and other reviewers comments, I have decided to increase my rating although there are still some minor problems: R2 analysis relies on some assumptions that are only valid in the linear case and can be deceptive in the nonlinear case [E]

[E] Spiess, Andrej-Nikolai, Natalie Neumeyer. An evaluation of R2 as an inadequate measure for nonlinear models in pharmacological and biochemical research: a Monte Carlo approach. BMC Pharmacology. 2010; 10: 6.

Also, the reference [C] is incorrectly attributed as a particle swarm algorithm. In my previous comment, I meant PS as in policy search, not particle swarm. Apologies for the confusion.

---

> ### Author Response · Authors · 2023-11-15
>
> > (1) It is never explained the type of policy being used. / The derivation is based on nonparametric (tabular) policies, but those type of policies are too limited for the MuJoCo experiments. / Even the linear policy seems to be very limited.
>
>
> We apologize for the possible lack of clarity in the paper that might have led to this confusion. We have updated the paper to clarify that our theoretical results apply to all kinds of policies.
>
> RL with deterministic linear policies is a standard benchmark for the Bayesian optimization setting [1,2,3]. Indeed, for computational considerations, BO cannot easily scale to the thousands of dimensions required for a neural network actor. At least, not with the standard algorithms.
>
> In our case, like in previous works, e.g., MPD and ARS, we show that linear policies are not too limiting for the MuJoCo experiments, as clearly evidenced by our successful experimental results.
>
> Our long-term goal is for Bayesian optimization to also scale to the larger neural network parameterizations of the policy, and we think that our proposed mean function that bridges a gap between BO and policy gradient methods is a good first step towards this goal.
>
> > (2) The idea of incorporating information from x at theta is interesting, but that still requires to sample rollouts from the policy pi_x, which seems counterintutive in a BO setting where sample efficiency is of paramount importance.
>
> We want to point out that this is a misunderstanding. Indeed, the returns and occupation measures of the policies $\pi_x$ used for the validation come from the already collected data. We make this point clearer in our update of Section 4. Thank you for your feedback.
>
> > (3) It seems that the philosophy behind the mean function is similar to having a GP with inducing points. The other advantage of using pi_x seems exploration, but that can also be achieved with importance sampling.
> Given the nature of the application (sample efficient PS), the authors should include in the comparison both novel AC methods (SAC, TQC…), model based approaches which are also sample efficient (MBPO…). For example, according to the MBPO paper [A], MBPO is able to achieve 6000 reward in a similar number of steps than ABS gets 4000.
>
>
>  We kindly point out that the aim of our paper is to construct an effective algorithm for RL that explores using only deterministic policies. As cited in our introduction, we consider the problem of continuous MDPs. As for SAC, MBPO, and other bootstrapping-based methods, they do not fit into our problem of performing exploration using only deterministic policies, as mentioned in our abstract.
>
>
> > (4) In the paragraph before eq 8, it says “Proposition 3.3”. I think you mean “Theorem 3.3”
>
> Thank you for pointing this out. We fixed the reference.

---

> ### Author Response · Authors · 2023-11-15
>
> > (5.a) Why all the policies in Figs 3 and 4 are normalized?
>
> > (5.b) What do you mean by discounted policy returns? It seems that the undiscounted returns like the ones presented in the appendix are more standard in the literature and allows comparisons with other works.
>
> In Figure 3,  we choose to report the discounted policy returns. Indeed, the RL agent is trying to optimize the discounted returns. Hence, we believed it would make more sense to report these. Since we report discounted returns, we believed that reporting the normalized values would allow for easier comparison. Finally, having noticed that certain policies have high discounted returns and a relatively lower undiscounted return (e.g. Hopper), we opted to report the undiscounted returns in the appendix in Figure (5) for more transparency and ease of comparison with other works.
>
>
> > (6) Also, the standard approach is to use steps for the horizontal axis, instead of episodes.
>
> We report the policy return as a function of the number of episodes because direct policy search methods require rollouts of entire episodes. The majority of the literature of BO applied in RL uses this metric as a measure of their empirical sample efficiency [1,2,3]. In our experiments, all the tasks have a fixed horizon of 1000 steps.
>
> > (7) The R2 analysis is unclear. The fact that for many iterations the correlation is negative and it is mostly below 0.5 means that the fitness might not be accurate. Furthermore, the R2 is a fitness measurement for linear regression, which does not seem to be the case for the Q-function.
>
> We kindly point out that the R2 factor is a general metric to quantify the goodness-of-fit in regression, and is not only restricted to linear regression.
>
> We do not claim that the Q-function is always accurate. On the contrary, our results show large fluctuations in the validation metric at times.
>
> However, the validation metric has a median of over $50 \%$ for all the tasks. This means that the Q-function approximators are effective in reconstructing the returns of the last 3 batches of acquisition points that we choose to validate on.
>
> As for the test R2 score, it is as well consistently positive with medians of $> 50 \%$ for the "Ant" and "Halfcheetah" tasks.
>
> For "Walker2d", it seems to be harder to get a good test metric as the test R2 score has a median of $-20\%$. This is probably due to the lack of smoothness of the "Walker2d" making the residual terms higher, and thus, harder for our mean function to fit even with a perfect Q-function estimator. However, despite this, we see a large increase in the performance in "Walker2d" compared to the MPD baseline proving that we do not need the mean function to be a perfect predictor of the whole neighborhood but we need it just to identify a few good search directions.
>
> To make the comprehension of results in Figure (2) easier, we make the following modifications:
>
> (a) We remove the averaging that was used to make the plots more readable and report the histograms of the distributions of the scores in the Appendix.
> (b) We report the distributions and median of the scores in Figure (6) in the appendix.
>
> > (8) Policy search with Bayesian optimization has been extensively studied since 2007 [B] and there are many works missing. For example, [C] presented a PS method with BO using local models as well, while [D] also combines parametric and non-parametric (GP) models.
>
> Thanks for the references. We have added the citations.
>
> [1] Müller, Sarah, Alexander von Rohr, and Sebastian Trimpe. "Local policy search with Bayesian optimization." Advances in Neural Information Processing Systems 34 (2021): 20708-20720.
>
> [2] Nguyen, Quan, et al. "Local Bayesian optimization via maximizing probability of descent." Advances in neural information processing systems 35 (2022): 13190-13202.
>
> [3] Eriksson, David, et al. "Scalable global optimization via local Bayesian optimization." Advances in neural information processing systems 32 (2019).

---

### Author Response · Authors · 2023-11-15
**Comments on the revised version**

We thank all Reviewers for their valuable feedback. We have uploaded a revised draft in which we make some changes to address the concerns and misunderstandings that have been raised.

As a summary, we have:

- Added a paragraph on the role of Corollary 3.3.1 which gives a better intuition of the interplay between the GP and the Q-function approximator.

- Improved Section 4 to make the proposed validation and test metrics clearer.

- Removed the smoothing used in the initial version of Figure 2, and added a histogram of the distribution of validation and test scores in the appendix to make the interpretation of our results easier.

- Added a summary of the results at the end of Section 5.2.

- Fixed the various typos and clarified several claims following the reviewers' feedback.

- We have updated the figures to improve readability (the results are unchanged).

---

### Meta-Review · Area_Chair_WrYe · 2023-12-06

**Metareview:**

The paper combines Bayesian Optimisation with Deterministic Policy Gradient. The proposed method explores with deterministic policies. The Bayesian model jointly models the value function and the gradient. Experiments are limited to linear policies.

Strengths:
- interesting combination of BO and PG, which try to leverage the benefits of both approaches
- experiments are sufficient to prove the main point the paper is trying to make

Weaknesses:
- use of linear policies somewhat limiting (if undertstandable)
- an extension to risk-sensitive goals would be nice (but is not a requirement for acceptance)
- lack of a huge experimental win (which is fine in a paper of this type).

**Justification For Why Not Higher Score:**

Paper is somewhat marginal & incremental  (see list of weaknesses in the meta-review).

**Justification For Why Not Lower Score:**

Paper addresses an important problem well and deserves a place in the ICLR community (see list of strengths in the meta-review).

---

### Decision · Program_Chairs · 2024-01-16

Accept (poster)